# On Privacy and Personalization in Cross-Silo Federated Learning

**Ziyu Liu**   **Shengyuan Hu**   **Zhiwei Steven Wu**   **Virginia Smith**
Carnegie Mellon University
{kzliu, shengyuanhu, zstevenwu, smithv}@cmu.edu

## Abstract

While the application of differential privacy (DP) has been well-studied in cross-device federated learning (FL), there is a lack of work considering DP and its implications for cross-silo FL, a setting characterized by a limited number of clients each containing many data subjects. In cross-silo FL, usual notions of *client-level* DP are less suitable as real-world privacy regulations typically concern the in-silo data subjects rather than the silos themselves. In this work, we instead consider an alternative notion of *silo-specific sample-level* DP, where silos set their own privacy targets for their local examples. Under this setting, we reconsider the roles of personalization in federated learning. In particular, we show that mean-regularized multi-task learning (MR-MTL), a simple personalization framework, is a strong baseline for cross-silo FL: under stronger privacy requirements, silos are incentivized to federate more with each other to mitigate DP noise, resulting in consistent improvements relative to standard baseline methods. We provide an empirical study of competing methods as well as a theoretical characterization of MR-MTL for mean estimation, highlighting the interplay between privacy and cross-silo data heterogeneity. Our work serves to establish baselines for private cross-silo FL as well as identify key directions of future work in this area.

## 1 Introduction

Recent advances in machine learning often rely on large, centralized datasets [84, 23, 65], but curating such data may not always be viable, particularly when the data contains private information and must remain siloed across clients (e.g. mobile devices or hospitals). Recently, federated learning (FL) [71, 50] has emerged as a paradigm for learning from such distributed data, but it has been shown that its data minimization principle alone may not provide adequate privacy protection for participants [101, 102]. To obtain formal privacy guarantees, there has thus been extensive work applying *differential privacy* (DP) [27, 28] to various parts of the FL pipeline (e.g. [32, 73, 41, 48, 4, 63, 44, 83, 34]).

Existing approaches for differentially private FL are typically designed for *client-level* DP in that they protect the federated clients, such as mobile devices ("user-level"), tasks in multi-task learning ("task-level"), or data silos like institutions ("silo-level"), and DP is achieved by clipping and noising the client model updates. While client-level DP is considered a strong privacy notion as all data of a single client is protected, it may not be suitable for *cross-silo* FL, where there are fewer clients but each hold many data subjects that require protection. For example, when hospitals/banks/schools wish to federate patient/customer/student records, it is the people owning those records rather than the participating silos that should be protected. In fact, laws and regulations may mandate such participation in FL be disclosed publicly [96], compromising the privacy of the federating clients.

In this work, we instead consider a more natural model of *silo-specific sample-level privacy* (Fig. 1, with variants appearing in [41, 66, 109, 51]): the $k$-th silo may set its own $(\varepsilon_k, \delta_k)$ sample-level DP target for any learning algorithm with respect to its local dataset. With this formulation in mind, we then reconsider the impact of privacy, heterogeneity, and personalization in cross-silo FL. In particular,

36th Conference on Neural Information Processing Systems (NeurIPS 2022).

we explore existing baselines for FL (mostly developed in cross-device settings) across private cross-silo benchmarks, and we find that the simple baseline of mean-regularized MTL (MR-MTL) has many advantages for this setting relative to other more common (and possibly more complex) methods. We then further analyze the performance of MR-MTL under varying levels of heterogeneity and privacy, both in theory and practice. In addition to establishing baselines for cross-silo FL, we also identify interesting future directions in this area (§7 and Appendix G). We summarize our contributions below:[1]

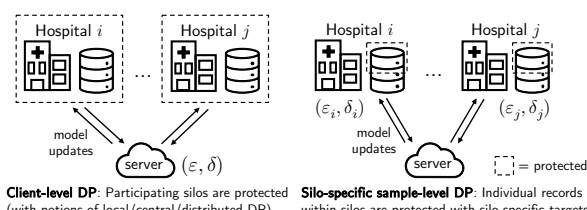

Figure 1: **Client-level** DP vs **Silo-specific sample-level** DP.

- We consider the notion of *silo-specific sample-level differential privacy* (DP) as a more realistic privacy model for *cross-silo federated learning* (FL). We analyze its implications on existing FL algorithms and, in particular, how it interfaces with data heterogeneity across silos.

- We empirically show that mean-regularized multi-task learning (MR-MTL), a simple form of model personalization, is a remarkably strong baseline under silo-specific sample-level DP. Core to its effectiveness is its ability to (roughly) interpolate on the model personalization spectrum between local training and FedAvg with minimal privacy overhead.

- We provide a theoretical analysis of MR-MTL under mean estimation and characterize how MR-MTL navigates the tension between privacy and cross-silo data heterogeneity.

- Finally, we examine the complications of deploying an optimal MR-MTL instance that stem from the privacy cost of *hyperparameter tuning*. Our reasoning also applies to other personalization methods whose advantage over local training and/or FedAvg hinges on selecting the best hyperparameter(s). This raises important questions around the practicality of leveraging personalization to balance the emerging tradeoffs under silo-specific sample-level DP.

## 2 Preliminaries

**Federated Learning (FL)** [71, 58, 50] is a distributed learning paradigm with an emphasis on data protection: in every training round, each client (participant) downloads the current global model from a central server, trains it with the local dataset, and uploads the model changes (instead of the data) back to the server, which then aggregates the changes into a new global model. A basic instantiation of FL is FedAvg [71], where clients are stateless and the server performs a simple (weighted) average. *Cross-device* FL refers to settings with many clients each with limited data, bandwidth, availability, etc. (e.g. mobile devices). In contrast, *cross-silo* FL typically involves less clients (e.g. banks, schools, hospitals) but each with more resources. Two distinguishing characteristics of cross-silo FL relevant to our work are that (1) silos may have sufficient data to fit a reasonable local model *without* FL, and (2) each data point in a silo tends to map to a data subject (a person) requiring privacy protection.

**Differential Privacy (DP).** Despite its ability to mitigate systemic privacy risks, FL by itself does not provide formal privacy guarantees for participants' data [50, 102, 101], and differential privacy is often used in conjunction with FL to ensure that an algorithm does not leak the privacy of its inputs.

**Definition 2.1** (Differential Privacy [27, 28]). A randomized algorithm $M : \mathcal{X}^n \to \mathcal{Y}$, where $\mathcal{X}^n$ is the set of datasets with $n$ samples and $\mathcal{Y}$ is the set of outputs, is $(\varepsilon, \delta)$-DP if for any subset $S \subseteq \mathcal{Y}$ and any neighboring $x, x'$ differing in only one sample (by replacement), we have

$$\Pr[M(x) \in S] \le \exp(\varepsilon) \cdot \Pr[M(x') \in S] + \delta. \tag{1}$$

To apply DP to a dataset query, one commonly used method is the Gaussian mechanism [28], which involves bounding the contribution ($\ell^2$-norm) of each sample in the dataset followed by adding Gaussian noise proportional to that bound onto the aggregate. To apply DP in FL, one needs to define the "dataset" to protect; typically, as in *client-level* DP, this is the set of FL participants and thus the model updates from each participant in every round should be bounded and noised. In learning settings, we need to repeatedly query a dataset and the privacy guarantee composes. We use DP-SGD [92, 11, 1] for ensuring sample-level DP for model training, and we use Rényi DP [77] and zCDP [15] for tight privacy composition. In certain FL algorithms, clients also perform additional

---

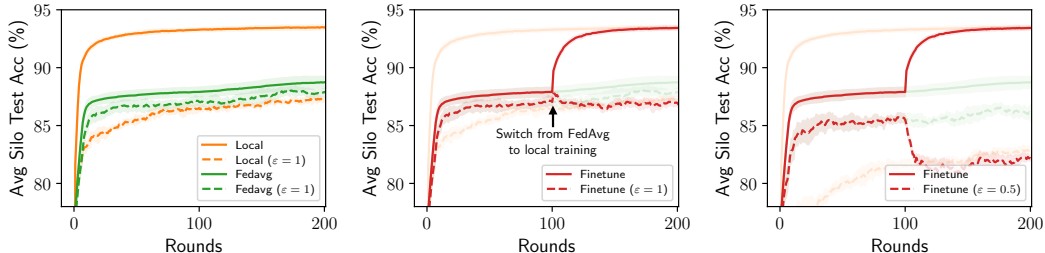

Figure 2: **Two notable phenomena under *silo-specific sample-level DP***: (1) FedAvg can serve to cancel out per-silo DP noise and thus outperform local training even when the latter works better without privacy (left); (2) Local finetuning [104, 19] (FedAvg followed by local training) may not improve utility as expected, as the effect of noise reduction is removed when finetuning begins (mid & right). Results report mean test acc $\pm$ std on the Vehicle dataset over 5 seeds. For simplicity, all silos budgets for the same labeled $\varepsilon$ with $\delta = 10^{-7}$. Transparent curves refer to local/FedAvg runs with the same $\varepsilon$ labeled for finetuning (compare left & mid).

work such as cluster selection [69, 33] that incurs privacy overhead with respect to its local dataset that must be accounted for independently from DP-SGD. See Appendix A for additional background.

**Personalized FL.** Model personalization is a key technique for improving utility under data heterogeneity across silos.[2] Past work has examined the roles of local adaptation [99, 104, 19], multi-task learning [91, 89], clustering [33, 22, 69, 89], public data [108, 69], meta learning [46, 56, 31], or other forms of model mixtures [61, 57, 69, 38, 24, 3]. Notably, many methods leverage extra computation to some extent (e.g. extra iterations [61, 57, 19] or cluster selection [33, 69]), which will result in privacy overhead under silo-specific sample-level DP as discussed in the following section. Of particular interest is the family of mean-regularized multi-task learning (MR-MTL) methods [30, 94, 38, 37] (see Algorithm A1 for a typical instantiation). We find that MR-MTL, while extremely simple, is a strong baseline for private cross-silo FL.

## 3 Revisiting the Privacy Model for Cross-Silo Federated Learning

To date, the prevalent privacy model for federated learning has been to protect the participating clients, i.e. client-level DP. For cross-silo FL, however, several factors render client-level DP less appropriate. First, cross-silo FL often involves a small number of clients and it can be utility-wise more costly to attain the same privacy targets. For example, privacy amplification via sampling [1, 78] may not apply on the client level since all silos typically participate in every round. Second, many existing methods focus on enforcing client-level DP in a non-local model and thus defines a shared privacy target for all participants, but in real-world cross-silo settings, participants under different jurisdictions (e.g. states) may have varying privacy requirements and thus opt for different privacy-utility tradeoffs. Third, while silo-level protection implies sample-level protection, it may be too stringent in practice as silos often have large local datasets. These unique properties for private cross-silo learning motivate us to consider *silo-specific sample-level* DP as an alternative privacy model (Fig. 1):

**Definition 3.1** (Silo-specific sample-level DP). A cross-silo FL algorithm with $K$ clients (silos) satisfy $\{(\varepsilon_k, \delta_k)\}_{k \in [K]}$-"*silo-specific sample-level DP*" if the local (personalized) model $M_k$ of every silo $k \in [K]$ satisfies $(\varepsilon_k, \delta_k)$-DP w.r.t. the silo's local dataset of training examples.

**Characteristics of silo-specific sample-level privacy.** Importantly, silo-specific sample-level DP is defined over the *disjoint* datasets of the individual silos, rather than the combined dataset of all silos.[3] To instantiate this setup in FL, each silo can simply run DP-SGD [92, 11, 1] with a noise scale calibrated to gradually spend its privacy budget over $T$ training rounds, and return the noisy model update at each round. This privacy notion has several important implications on the dynamics of FL:

1. **Silos incur privacy costs with queries to their data, but *not* with participation in FL.** This follows from DP's robustness to post-processing: the silos' model updates in each round already satisfy their own sample-level DP targets, and participation by itself does not involve extra dataset

---

[2]Note that "personalization" refers to customizing models for each *client* in FL rather than a specific person.

[3] A record in such a combined dataset is at most $(\max_i \varepsilon_i, \max_i \delta_i)$-DP [76, 103, 63]. Moreover, if multiple records (either within a silo or across silos) map to the same person, then it is more intricate to protect the person rather than their records. Here we focus on the case where each entity has at most one record across the combined dataset (e.g. students attending exactly one school). See Appendix B for discussions.

queries (e.g. DP-SGD steps). In contrast, local training without communication can be kept noise-free under client-level DP, but participation in FL requires privatization. Two immediate consequences of the above are that (1) local training and FedAvg now have *identical* privacy costs, and (2) *local finetuning* for model personalization may no longer work as expected (Fig. 2).

2. **Less reliance on a trusted server.** As a corollary of the above, all model updates of silo $k$ satisfy (at least) $(\varepsilon_k, \delta_k)$-DP against external adversaries, including all other silos and the orchestrating server [28, 109]. In contrast, client-level DP under a non-local model necessitates some trust on the server, even for distributed DP methods (e.g. [29, 21, 48, 4, 20, 18]).

3. **Tradeoff emerges between costs from privacy and heterogeneity.** As privacy-perserving noises are added independently on each silo, they are reflected in silos' model updates and can thus be mitigated when the model updates are aggregated (e.g. via FedAvg), leading to a smaller utility drop due to DP for the shared model. On the other hand, federation also means that the shared model may suffer from *client heterogeneity* (non-iid data across silos). This intuition is observed in Fig. 2: while local training may outperform FedAvg without privacy (as a result of heterogeneity), the opposite can be true when privacy is added (as a result of noise variance reduction).

The first and last in the above are of particular interest because they suggest that *model personalization* can play a key and distinct role in our privacy setting. Specifically, local training (no FL participation) and FedAvg (full FL participation) can be viewed as two ends of a *personalization spectrum* with identical privacy costs; if local training minimizes the effect of data heterogeneity but enjoys no DP noise reduction, and contrarily for FedAvg, it is then natural to ask whether there exist personalization methods that lie in between and achieve better utility, and, if so, what methods would work best.

**Related privacy settings.** Past work on differentially private FL has concentrated on client-level DP and cross-device FL (e.g. [32, 74, 44, 34, 49, 48, 6]), and the application of DP in cross-silo FL, particularly where each silo defines its own DP targets for records of its own dataset, is relatively underexplored. Privacy notions closest to ours first appeared in [95, 56, 41, 109, 66, 51, 63]. In [95], each client adds its own one-shot noise onto its outgoing update, but in learning scenarios this provides client-level protection. The works of [56, 66, 109, 41, 63] study analogous privacy notions, though they respectively focus on boosting utility [56], analyzing statistical rates [66], adapting FL to $f$-DP [109, 25], applying security primitives [41], and learning a better global model; the aspects of heterogeneity, DP noise reduction, the personalization spectrum, and their interplay (e.g. Figs. 2 and 5) were unexplored. The work of [51] also considers a similar privacy notion, but the authors study a disparate trust assumption where DP noise is *not* added to local training/finetuning such that the final personalized models lack privacy guarantees. We note that the trust model most suitable for private cross-silo FL may be application-specific; in this work, we focus on the setting where the outputs of the FL procedure (the personalized models) must remain differentially private.

## 4   Baselines for Private Cross-Silo Federated Learning

With the characteristics from §3 in mind, we now explore various methods on cross-silo benchmarks. We defer additional details as well as results on more settings and datasets to the appendix.

**Datasets.** We consider four cross-silo datasets that span regression/classification and convex/non-convex tasks: Vehicle [26], School [35], Google Glass (GLEAM) [82], and CIFAR-10 [53]. The first three datasets have real-world cross-silo characteristics: Vehicle contains measurements of road segments for classifying the type of passing vehicles, School contains student attributes for predicting exam scores, and GLEAM contains motion tracking data to classify wearers' activities. CIFAR-10 has heterogeneous client splits following [94, 90]. See Appendix C.1 for more details and datasets.

**Benchmark methods.** We consider several representative methods in the personalized FL literature beyond local training and FedAvg [71]: local finetuning [99, 104, 19] (a simple but strong baseline for model personalization), Ditto [57] (state-of-the-art personalization method), Mocha [91] (personalization with task relationship learning), IFCA/HypCluster [33, 69] (state-of-the-art hard clustering method for client models), and the mean-regularized multi-task learning (MR-MTL) methods [30, 94, 38, 37] (which we analyze in §5). For fair comparison under silo-specific sample-level DP, we align all benchmark methods on the total privacy budget by first restricting the total number of iterations over the local datasets and then account for any privacy overheads (in the form of necessary extra steps [57] or cluster selection for IFCA/HypCluster [33, 69]). Importantly, many other personalization methods can either be reduced to one of the above under convex settings (e.g. [46, 61]) or are unsuitable due to large privacy overheads (e.g. large factor of extra steps for [31]).

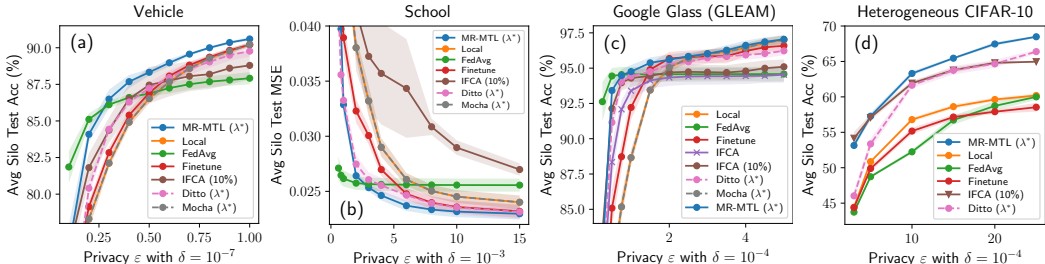

Figure 3: **Privacy-utility tradeoffs** (privacy budgets $\varepsilon$ vs. test metrics, mean $\pm$ std w/ 5 seeds) for various personalization methods on **Vehicle**, **School**, **GLEAM**, and **Heterogeneous CIFAR-10** datasets respectively. For simplicity, every silo targets for the same $(\varepsilon, \delta)$ under silo-specific sample-level DP. $\lambda^*$ denotes a tuned regularization strength where applicable. "Local" denotes local training (clients train and keep their own models). "IFCA (10%)" denotes forming clusters for only first 10% of training rounds due to privacy overhead (Fig. 4).

**Training setup.** For all methods, we use minibatch DP-SGD in each silo to satisfy silo-specific sample-level privacy; while certain methods may have more efficient solvers (e.g. dual form for [91]), we want compatibility with DP-SGD as well as privacy amplification via sampling on the example level for tight accounting. For all experiments, silos train for 1 local epoch in every round (except for [57] which runs $\geq 2$ epochs). Hyperparameter tuning is done via grid search for all methods. Importantly, when comparing the benchmark methods, we do not account for the privacy cost of hyperparameter tuning in order to focus on their inherent privacy-utility tradeoff; we revisit this issue in §7. For simplicity, we use the same privacy budget for all silos, i.e. $(\varepsilon_i, \delta_i) = (\varepsilon_j, \delta_j)$ for all $i, j \in [K]$; note that this is not a restrictive assumption since the effect of having varying DP noise scales from different budgets can be attained by varying local dataset sizes.

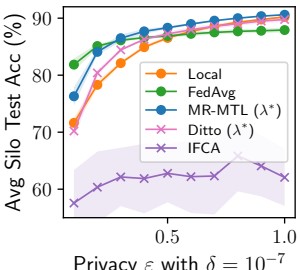

Figure 4: The privacy overhead of cluster selection (IFCA) and extra iterations (Ditto) compared to local, FedAvg, and MR-MTL under silo-specific sample-level DP.

**Results.** Fig. 3 shows the privacy-utility tradeoffs across four datasets. We observe that MR-MTL consistently outperforms a suite of baseline methods, and that it performs at least as good as local training and FedAvg (endpoints of the personalization spectrum), except at high-privacy regimes (possibly different for each dataset). In particular, there exists a range of $\varepsilon$ values where MR-MTL can give significantly better utility over local training and FedAvg *under the same privacy budgets* ($\varepsilon \approx 0.5, 6, 1.5$ for Fig. 3 (a, b, c) respectively); this is our key regime of interest.

**Effects of silo-specific sample-level privacy.** In Fig. 2 we saw that local finetuning may not improve utility as expected, motivating us to reconsider the roles of federation and personalization (§5 below). In Fig. 4, we consider the implication of silo-specific sample-level DP from the effects of privacy overhead due to additional dataset queries: (1) If IFCA [33, 69] performs cluster selection at every round (default behavior), then the extra privacy cost can be prohibitive; (2) Despite its similarity to MR-MTL, Ditto [57]'s privacy overhead makes it less competitive (see also Fig. 5).

## 5 On the Effectiveness of Mean-Regularized MTL for Private Cross-Silo FL

Following the observations in §4, we now examine the desirable properties of a good algorithm under silo-specific sample-level privacy and understand why MR-MTL may be an attractive candidate.

**Federation as noise reduction.** A key message from §3 and Fig. 2 is that the utility cost from DP can be significantly smaller for FedAvg compared to local training even when the latter spends an identical privacy budget. This implies that FedAvg may have inherent benefits for DP noise reduction, despite the noise are added in the *gradient* space rather than the parameter space (as in client-level DP). Consider a simple setting of DP gradient descent: the update rule $w_k^{(t+1)} = w_k^{(t)} - \frac{\eta}{n_k} \left( z^{(t)} + \sum_{i=1}^{n_k} g_{k,i}^{(t)} \right)$ for silo $k$ recursively expands to $w_k^{(T)} = w_k^{(0)} - \frac{\eta}{n_k} \sum_{t=0}^{T-1} z^{(t)} - \frac{\eta}{n_k} \sum_{t=0}^{T-1} \sum_{i=1}^{n_k} g_{k,i}^{(t)}$ over $T$ steps, where $g_{k,i}^{(t)}$ is the clipped gradient of the $i$-th local example at step $t$ (with norm bound $c$) out of a total of $n_k$ examples, and $z^{(t)} \sim \mathcal{N}(0, \sigma^2 \mathbf{I})$ is the Gaussian noise added to the gradient sum at step $t$ that targets for an overall privacy budget of $(\varepsilon_k, \delta_k)$ over $T$ steps

with $\sigma^2 = O\left(c^2 T \ln(1/\delta_k)/\varepsilon_k^2\right)$ [1]. The cumulative noise term

$$Z^{(T)} \triangleq -\frac{\eta}{n_k} \sum_t z^{(t)} \sim \mathcal{N}\left(0, \; O\left(\frac{\eta^2 c^2 T^2 \ln(1/\delta_k)}{n_k^2 \varepsilon_k^2}\right) \cdot \mathbf{I}\right) \tag{2}$$

indeed implies that each silo's model update has an independent Gaussian random walk component [88, 7, 100] whose variance can be reduced by averaging with other silos' updates, as in FedAvg.[4] A similar reasoning applies to SGD cases since the additive DP noises are i.i.d. across the minibatches.

**Model personalization for privacy-heterogeneity cost tradeoff.** A major downside of FedAvg is that it may underperform simple local training due to data heterogeneity (e.g. [104] and Fig. 2), particularly given that clients in cross-silo FL often have sufficient data to fit reasonable local models. This suggests an emerging role for model personalization on top of its benefits in terms of utility [99], robustness [104], or fairness [57] under heterogeneity: our privacy model allows local training and FedAvg to be viewed as two endpoints of a *personalization spectrum* that respectively mitigate the utility costs of heterogeneity and privacy noise with identical privacy budgets (recall §3); this means that personalization methods could be viewed as interpolating between these endpoints and that various personalization methods essentially do so in different ways. However, our empirical observations motivate the following key properties of a good personalization algorithm:

1. **Noise reduction**: The effect of noise reduction is present throughout training so that the utility costs from DP can be consistently mitigated. Local finetuning is a counter-example (Fig. 2).
2. **Minimal privacy overhead**: There are little to no additional local dataset queries to prevent extra noise for DP-SGD under a fixed privacy budget. In effect, such privacy overhead can shift the utility tradeoff curve downwards, and Ditto [57] may be viewed as a counter-example (Fig. 4).
3. **Smooth interpolation along the personalization spectrum**: The interpolation between local training and FedAvg should be fine-grained (if not continuous) such that an optimal tradeoff should be attainable. Clustering [33, 69] may be viewed as a counter-example when there are no clear heterogeneity strucutre across clients.

These properties are rather restrictive and they render many promising algorithms less attractive. For example, model mixture [61, 13, 24] and local adaptation [104, 19] methods can incur linear overhead in dataset iterations, and so can multi-task learning [57, 91, 89] methods that benefit from additional training. Clustering methods [33, 22, 69, 89] can also incur overhead with cluster selection [33, 69], distillation [22], or training restarts [89], and they discretize the personalization spectrum in a way that depends on external parameters (e.g., the number of clients, clusters, or top-down partitions).

**The case for mean-regularization.** These considerations point to mean-regularized multi-task learning (MR-MTL) as one of the simplest yet particularly suitable forms of personalization. MR-MTL has manifested in various forms in the literature [30, 105, 94, 38, 19, 44] with the key idea that a personalized model $w_k$ for each silo $k$ should be close to the mean of all personalized models $\bar{w}$ via a regularization penalty $\lambda/2\|w_k - \bar{w}\|_2^2$ (see Algorithm A1 for a typical instantiation). The hyperparameter $\lambda$ serves as a smooth knob between local training and FedAvg, with $\lambda = 0$ recovering local training and a larger $\lambda$ forces the personalized models $w_k$ to be closer to each other ("federate more"). However, it is an imperfect knob as $\lambda \to \infty$ may *not* recover FedAvg under a typical optimization setup as the regularization term may dominate the gradient step $w_k^{(t+1)} = w_k^{(t)} - \eta\left(g_t + \lambda\left(w_k^{(t)} - \bar{w}^{(t)}\right)\right)$ where $g_t$ is the noisy clipped gradient, and MR-MTL may thus underperform FedAvg in high-privacy regimes that necessitate a large $\lambda$ to mitigate DP noise (Fig. 3).

MR-MTL has the attractive properties that: (1) noise reduction is achieved throughout training via a soft constraint that personalized models are close to an averaged model; (2) for fixed $\lambda$ it has *zero* additional privacy cost compared to local training/FedAvg as it does not involve extra dataset queries; and (3) $\lambda$ provides a smooth interpolation along the personalization spectrum. Moreover, compared to other regularization-based MTL methods, it adds only one hyperparameter $\lambda$ (cf. [45, 110, 36]); this has important practical implications as will be discussed in §7. It also has fast convergence [106] and easily extends to deep learning with good empirical performance in the primal [94, 44] (cf. [10, 91, 64]). It is also sufficently extensible to handle structured heterogeneity (discussed below).

---

[4] The work of [49] examines the benefits of adding negatively correlated (instead of independent) noises $z_t$ across time steps. While this is a potential orthogonal extension to our use of local DP-SGD *within* each silo, it may not be directly applicable to our main focus of reducing noise variance *across* silos, since for each silo $k$ to satisfy its own $(\varepsilon_k, \delta_k)$ requirement, it must add noise independent to other silos.

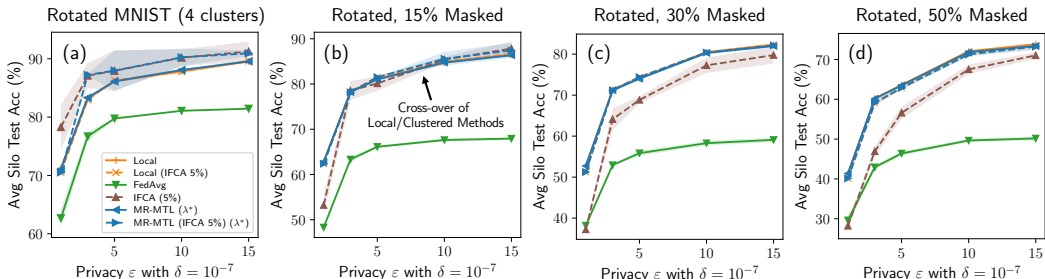

Figure 6: **Test acc ± std on Rotated & Masked MNIST**. (IFCA 5%) denotes warm-starting the method by running IFCA [33, 69] for first 5% of rounds followed by running the method within the cluster structures.

We argue through the following empirical analyses that these properties make MR-MTL a strong baseline under silo-specific sample-level DP.

**Navigating the emerging privacy-heterogeneity cost tradeoff.** In Fig. 5 we study the effect of the regularization strength $\lambda$ on the model utility directly. There are several notable observations: (1) In both private and non-private settings, $\lambda$ serves to roughly interpolate between local training and FedAvg. (2) The utility at the best $\lambda^*$ may outperform both endpoints. This is significant for the private setting since MR-MTL achieves an *identical* privacy guarantee as the endpoints. (3) Moreover, the *advantage* of MR-MTL over the best of the endpoints are also larger under privacy. (4) The value of $\lambda^*$ also increases under privacy, indicating that the personalized silo models are closer to each other (i.e. silos are encouraged to "federate more") for noise reduction. We will characterize these behaviors in §6. We also consider Ditto [57] in Fig. 5,

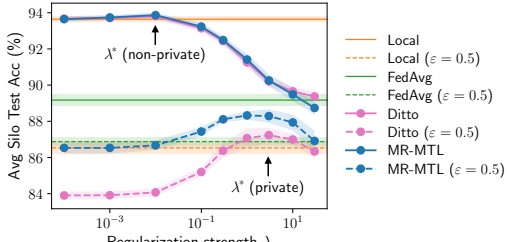

Figure 5: **Test acc ± std of MR-MTL** with varying $\lambda$ (corresponding to $\varepsilon = 0.5$ in Fig. 3 (a)). Optimal points ($\lambda^*$) exist where it outperforms both ends of the spectrum under the *same* privacy. Ditto [57] gives a similar interpolation but has strictly worse privacy-utility tradeoffs due to its privacy overhead. See Appendix H.2 for extensions of this figure to other privacy settings and datasets.

a state-of-the-art personalization method that resembles MR-MTL and exhibits similar behaviors, to illustrate the effect of privacy overhead from its extra local training iterations.

**MR-MTL under structured heterogeneity.** We further study (1) the extensibility of MR-MTL as a strong baseline method to handle clustering structures of silo data distributions and (2) its flexibility to handle varying heterogeneity levels, by manually introducing two layers of heterogeneity to the MNIST dataset [54]. The first layer is 4-way rotations: train/test images are evenly split into 4 groups of 10 silos, with each group applying $\{0°, 90°, 180°, 270°\}$ of rotation to their images. The second layer is *silo-specific* masking: each silo generates and applies its unique random mask of $2 \times 2$ white patches to its images, with varying masking probability. Together, the 1st layer creates 4 well-defined silo clusters, and the 2nd layer (gradually) adds *intra-cluster* heterogeneity. Importantly, our goal is not to contrive a utility advantage of MR-MTL (in fact, the added heterogeneity is disadvantageous to mean-regularization), but to examine its extensibility and flexibility as a strong baseline method to match the best methods by construction. Under the 1st layer of heterogeneity, clustered FL methods [33, 69] should be optimal if the correct clusters are formed since there is no intra-cluster heterogeneity; with increasing silo-specific heterogeneity in the 2nd layer, local training should be increasingly more attractive. See Appendix C.1 for more details on the setup and examples of images.

We propose a simple heuristic to precondition or "warm-start" MR-MTL with a small number of training rounds by running private clustering (with IFCA [33, 69]) followed by mean-regularized training *within each formed cluster* (see Appendix D for details). We find that this simple heuristic, with convergence properties carried forward from its components [33, 30, 106], enables MR-MTL to excel at all levels of heterogeneity: in Fig. 6 (a), the preconditioning allows MR-MTL to match IFCA (optimal by construction) while local training (full personalization) does not benefit from the same preconditioning; in Fig. 6 (b, c, d), MR-MTL remains optimal across different levels of silo-specific heterogeneity (the 2nd layer) while the gains from warm-start gradually drop. We argue that extensibility and flexibility are good properties that make MR-MTL a strong baseline, as heterogeneity in practical settings is likely less adversarial than what we presented.

# 6 Analysis

In this section we provide a theoretical analysis of MR-MTL under mean estimation as a simplified proxy for (single-round) FL using a Bayesian framework extending on [57]. We provide expressions for the Bayes optimal estimator MR-MTL ($\lambda^*$) and describe how MR-MTL behaves with varying $\lambda$ in relation to the personalization spectrum to characterize our observations from Fig. 5. Proofs and extensions are deferred to Appendix E.

**Setup.** We start with a total of $K$ silos where the $k$-th silo holds $n$ training samples $X_k \triangleq \{x_{k,i} \in \mathbb{R}\}_{i \in [n]}$, each normally distributed around a hidden center $w_k$ with variance $\sigma^2$; i.e. $x_{k,i} = w_k + z_{k,i}$ with $z_{k,i} \sim \mathcal{N}(0, \sigma^2)$. To quantify heterogeneity, the silo centers $\{w_k\}_{k \in [K]}$ are also normally distributed around some unknown fixed meta-center $\theta$ with variance $\tau^2$; i.e. $w_k = \theta + z$ with $z \sim \mathcal{N}(0, \tau^2)$. A large $\tau$ means that the silo centers are distant from each other and thus their local objectives are heterogeneous, and contrarily for a small $\tau$. Our goal is for each silo $k$ to compute a sample-level private estimate of $w_k$ that minimizes the *generalization* error (i.e. on unseen points from the same local distribution). Each silo targets $(\varepsilon, \delta)$ sample-level DP and runs the Gaussian mechanism with noise scale $\sigma_{\mathrm{DP}} = c\sqrt{2 \ln(1.25/\delta)}/\varepsilon$ and clipping bound $c$.[5] Under this setting, the MR-MTL objective for the $k$-th silo is

$$h_k(w) = \tilde{F}_k(w) + \frac{\lambda}{2} \|w - \bar{w}\|_2^2. \tag{3}$$

Here, $\tilde{F}_k(w) \triangleq \frac{1}{2}(w - \frac{1}{n}(\xi_k + \sum_{i=1}^n x_{k,i} \cdot \min(1, c/\|x_{k,i}\|_2)))^2$ is the local objective to privately estimate the mean of the local data points with privacy noise $\xi_k \sim \mathcal{N}(0, \sigma_{\mathrm{DP}}^2)$. Since the data are (sub-)Gaussian, we assume one can choose $c$ such that no clipping error is introduced w.h.p., so $\hat{w}_k \triangleq \operatorname{argmin} \tilde{F}_k(w) = \frac{1}{n}(\xi_k + \sum_i x_{k,i})$ is the best local estimator. $\bar{w} = \frac{1}{K} \sum_k \hat{w}_k$ is the average estimator across silos, which is the same as the FedAvg estimator under mean estimation. We also consider the *external* average local estimators for silo $k$, defined as $\hat{w}_{\setminus k} \triangleq \frac{1}{K-1} \sum_{j \neq k} \hat{w}_j$. The following lemma gives the best MR-MTL estimator $\hat{w}_k(\lambda)$ as a function of $\lambda$.

**Lemma 6.1.** *Let $\lambda \geq 0$ and $\alpha = \frac{K+\lambda}{(1+\lambda)K} \in (1/K, 1]$. The minimizer of $h_k(w)$ is given by*
$$\hat{w}_k(\lambda) = \alpha \cdot \hat{w}_k + (1-\alpha) \cdot \hat{w}_{\setminus k}. \tag{4}$$

Note that the best $\lambda$ is always 0 for *training* error (i.e. estimating the empirical mean of the local data $\{x_{k,i}\}$); our hope is that with some $\lambda > 0$, $\hat{w}_k(\lambda)$ yields a better *generalization* error.

We now present the main takeaways. At a high level, the basis of our analysis relies on expressing the true center $w_k$ in terms of $\hat{w}_k$ and $\hat{w}_{\setminus k}$ conditioned on the local datasets $\{X_k\}_{k \in [K]}$. Let

$$\sigma_{\mathrm{loc}}^2 \triangleq \frac{\sigma^2}{n} + \frac{\sigma_{\mathrm{DP}}^2}{n^2} \tag{5}$$

denote the "local variance" of $\hat{w}_k$ around $w_k$ due to both data sampling and privacy noise.

**Behavior of MR-MTL at optimal $\lambda^*$.** We first derive the following lemma using Lemma 11 of [68].

**Lemma 6.2.** *Given $\hat{w}_k$, $\hat{w}_{\setminus k}$, and $\{X_k\}_{k \in [K]}$, we can express $w_k = \mu_k + \zeta_k$, where $\zeta_k \sim \mathcal{N}(0, \sigma_w^2)$,*

$$\sigma_w^2 \triangleq \left( \frac{1}{\sigma_{\mathrm{loc}}^2} + \frac{K-1}{K\tau^2 + \sigma_{\mathrm{loc}}^2} \right)^{-1} \quad and \quad \mu_k \triangleq \sigma_w^2 \left( \frac{1}{\sigma_{\mathrm{loc}}^2} \cdot \hat{w}_k + \frac{K-1}{K\tau^2 + \sigma_{\mathrm{loc}}^2} \cdot \hat{w}_{\setminus k} \right). \tag{6}$$

Lemma 6.2 expresses the unobserved true silo centers $w_k$ in terms of the (private) empirical estimators $\hat{w}_k$ and $\hat{w}_{\setminus k}$. This expression requires conditioning on the datasets $X_k$ as they form the Markov blankets of $\hat{w}_k$. Combining Lemma 6.1 and Lemma 6.2 gives the optimal $\lambda$.

**Theorem 6.3** (Optimal MR-MTL estimate). *The best $\lambda^*$ for the generalization error is given by*

$$\lambda^* = \operatorname*{argmin}_{\lambda} \mathbb{E}\left[ (w_k - \hat{w}_k(\lambda))^2 \mid \hat{w}_k, \hat{w}_{\setminus k}, \{X_k\}_{k \in [K]} \right] = \frac{1}{n\tau^2} \left( \sigma^2 + \frac{\sigma_{\mathrm{DP}}^2}{n} \right). \tag{7}$$

Theorem 6.3 suggests that there indeed exists an optimal point $\hat{w}(\lambda^*)$ on the personalization spectrum. Moreover, $\lambda^*$ grows smoothly with stronger privacy ($\sigma_{\mathrm{DP}}^2 \to \infty$) to encourage silos to "federate more" with others. This was empirically observed in Fig. 5. We now characterize the utility of $\hat{w}(\lambda^*)$.

---

[5] For simplicity, we start with the same $n, \sigma, \sigma_{\mathrm{DP}}$ for all silos and extend to silo-specific values in Appendix E.

**Corollary 6.4** (Optimal error with $\hat{w}(\lambda^*)$). *The MSE of the optimal estimator $\hat{w}(\lambda^*)$ is given by*

$$\mathcal{E}^* \triangleq \mathbb{E}\left[(w_k - \hat{w}_k(\lambda^*))^2 \mid \hat{w}_k, \hat{w}_{\setminus k}, \{X_k\}_{k \in [K]}\right] = \sigma_w^2 = \frac{\sigma_{\mathrm{loc}}^2(\sigma_{\mathrm{loc}}^2 + K\tau^2)}{K(\sigma_{\mathrm{loc}}^2 + \tau^2)}. \tag{8}$$

Note also that $\hat{w}(\lambda^*)$ is the MMSE estimator of $w_k$. Using Corollary 6.4, we can compare $\hat{w}_k(\lambda^*)$ against the endpoints of the personalization spectrum (local training and FedAvg) with the following propositions.

**Proposition 6.5** (Optimal error gap to local training). *Let $\mathcal{E}_{\mathrm{loc}} \triangleq \mathbb{E}\left[(w_k - \hat{w}_k)^2 \mid X_k\right] = \sigma_{\mathrm{loc}}^2$ be the error of the local estimate. Then, compared to the optimal estimator $\hat{w}(\lambda^*)$ (Corollary 6.4), the local estimator incurs an additional error of*

$$\Delta_{\mathrm{loc}} \triangleq \mathcal{E}_{\mathrm{loc}} - \mathcal{E}^* = \left(1 - \frac{1}{K}\right) \cdot \frac{\sigma_{\mathrm{loc}}^4}{\sigma_{\mathrm{loc}}^2 + \tau^2}. \tag{9}$$

**Proposition 6.6** (Optimal error gap to FedAvg). *Let $\mathcal{E}_{\mathrm{fed}} \triangleq \mathbb{E}\left[(w_k - \bar{w})^2 \mid \{X_k\}_{k \in [K]}\right]$ be the error under FedAvg. Then, compared to the optimal estimator $\hat{w}(\lambda^*)$ (Corollary 6.4), the FedAvg estimator incurs an additional error of*

$$\Delta_{\mathrm{fed}} \triangleq \mathcal{E}_{\mathrm{fed}} - \mathcal{E}^* = \left(1 - \frac{1}{K}\right) \cdot \frac{\tau^4}{\sigma_{\mathrm{loc}}^2 + \tau^2}. \tag{10}$$

Together, Propositions 6.5 and 6.6 suggest that the effects of stronger privacy ($\sigma_{\mathrm{DP}}^2, \sigma_{\mathrm{loc}}^2 \to \infty$) on how MR-MTL compares against the personalization endpoints are mixed, with the benefit of MR-MTL *increasing* against local training and *diminishing* against FedAvg. They also suggest that MR-MTL has an optimal utility advantage over both the endpoints when $\sigma_{\mathrm{loc}}^2 \approx \tau^2$ and local training performs on par with FedAvg, and the utility "bump" under privacy observed in Fig. 5 can be viewed as a result of this balance. It is worth noting that since the data variance $\sigma^2$ and heterogeneity $\tau^2$ are often fixed in practice, the freedom for silos to vary their privacy targets ($\varepsilon$ and $\sigma_{\mathrm{DP}}^2$) makes the utility advantage of MR-MTL more flexible compared to non-private settings.

**Behavior of MR-MTL as a function of $\lambda$.** The above captures how MR-MTL behaves at its optimum, but in Fig. 5 we also observed that MR-MTL has the desirable property that the utility cost from DP *shrinks smoothly* with larger $\lambda$ (§5). Lemma 6.7 and Theorem 6.8 below provides a characterization.

**Lemma 6.7** (Error of $\hat{w}_k(\lambda)$). *Let $\mathcal{E}(\lambda) \triangleq \mathbb{E}\left[(w_k - \hat{w}_k(\lambda))^2 \mid \hat{w}_k, \hat{w}_{\setminus k}, \{X_k\}_{k \in [K]}\right]$ be the error of MR-MTL as a function of $\lambda$. Then, $\mathcal{E}(\lambda) = \left(1 - \frac{1}{K}\right) \frac{\sigma_{\mathrm{loc}}^2 + \lambda^2\tau^2}{(\lambda+1)^2} + \frac{\sigma_{\mathrm{loc}}^2}{K}$.*

Using Lemma 6.7 we can now characterize how $\lambda$ affects the utility cost from DP (recall from Figs. 2 and 5 that federation helps with noise reduction). As a side note, Lemma 6.7 also suggests that MR-MTL's utility as a function of $\lambda$ would have a quasi-concave shape, as was empirically observed in Fig. 5. This could potentially help make heuristic or automated search over $\lambda$ easier.

**Theorem 6.8** (Private utility gap). *Let $\hat{w}_k(\lambda)$ and $\hat{w}_k^{\mathrm{DP}}(\lambda)$ be the non-private and private estimate of $w_k$ with $\sigma_{\mathrm{loc}}^2 \leftarrow \sigma^2/n$ and $\sigma_{\mathrm{loc}}^2 \leftarrow \sigma^2/n + \sigma_{\mathrm{DP}}^2/n^2$, respectively. Let $\mathcal{E}(\lambda)$ and $\mathcal{E}^{\mathrm{DP}}(\lambda)$ be the error of $\hat{w}_k(\lambda)$ and $\hat{w}_k^{\mathrm{DP}}(\lambda)$ respectively as in Lemma 6.7. Let $\Delta_{\mathrm{DP}}(\lambda) \triangleq \mathcal{E}^{\mathrm{DP}}(\lambda) - \mathcal{E}(\lambda)$ be the utility cost due to privacy as a function of $\lambda$. Then, $\Delta_{\mathrm{DP}}(\lambda) = \left(1 - \frac{1}{K}\right) \frac{1}{(\lambda+1)^2} \frac{\sigma_{\mathrm{DP}}^2}{n^2} + \frac{\sigma_{\mathrm{DP}}^2}{Kn^2}$.*

Theorem 6.8 suggests that with a larger $\lambda$, the utility cost from privacy can be smoothly mitigated by up to a factor of $K$, matching the empirical observation in Fig. 5.

## 7   Discussions

In previous sections, we empirically and theoretically studied the benefits of the best personalization hyperparameter $\lambda^*$ for MR-MTL, but it remains open as to how such $\lambda^*$ may be obtained. In this section, we take an honest look at the complications of deploying MR-MTL through the lens of the *privacy cost* of finding $\lambda^*$. There are in general several approaches: (1) a non-adaptive search (e.g. grid/random search [12]); (2) an adaptive search (e.g. grad student descent); or (3) an online estimation during training (e.g. [97, 8, 80]). Here, we focus on approach (1) since it is generic to all personalization methods and is a setting for which we have the best privacy accounting tools [62, 79] to our knowledge. We defer technical details and further discussions to Appendix F. Note that while we focus on MR-MTL, our reasoning in principle extends to all personalization methods whose advantage depends on having the best hyperparameter(s).

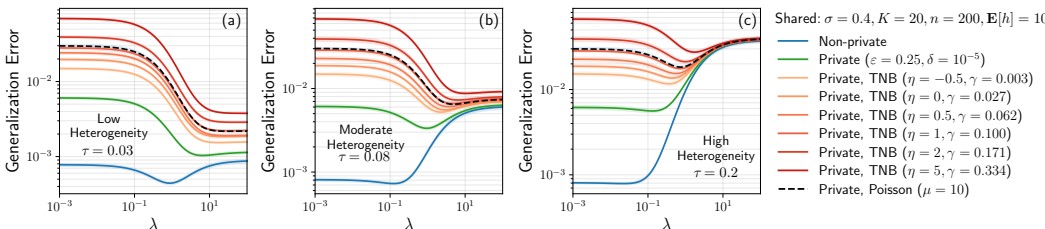

Figure 7: **Privacy costs of tuning $\lambda$ on mean estimation** (setup follows §6). Labels "**Private**" and "**Non-Private**" denote the errors of varying $\lambda$ with and without silo-specific sample-level DP (the privacy cost of tuning $\lambda$ is *not* included). "**Private, TNB/Poisson**" [79] denotes the same errors but accounts for the privacy cost of trying on average $\mathbf{E}[h] = 10$ values of $\lambda$, with $h$ sampled from the truncated negative binomial distribution with parameters $\eta, \gamma$ / the Poisson distribution with parameter $\mu$ to arrive at the same $\mathbf{E}[h]$. To interpret, observe that the lowest points of "**Private, TNB/Poisson**" may be still higher than one of the endpoints of "**Private**".

Recall that for a typical tuning procedure, a baseline algorithm $M$ is executed $h$ times with different hyperparameters and the best result is recorded. The work of [62, 79] shows that, with a constant $h$, there exists $M$ that satisfies $(\varepsilon, \delta = 0)$-DP where the output of tuning is not $(\tilde{\varepsilon}, 0)$-DP for any $\tilde{\varepsilon} < h\varepsilon$, with analogous negative results for Rényi DP (thus also for $\delta > 0$). This implies that naive tuning (as done in practice) can incur a prohibitive privacy overhead and obliterates the utility advantage of MR-MTL ($\lambda^*$) over local training/FedAvg. Instead, by making $h$ *random*, we can make $\tilde{\varepsilon}$ *constant* w.r.t. $h$ or at most $\tilde{\varepsilon} \leq O(\log \mathbf{E}[h])$ [62, 79]. However, using the simplified setting of mean estimation (§6), we find that even with this improved randomized protocol, there exist scenarios (Fig. 7) where the realistic cost of trying a moderate $\mathbf{E}[h] = 10$ values of $\lambda$ may significantly diminish, or even *outweigh*, the utility advantage of $\lambda^*$ over local training and FedAvg—that is, we might be better off by *not privately tuning $\lambda$ at all*.

The above has several important implications. On the negative side, it suggests that the true efficacy of MR-MTL can be smaller in practice. Moreover, it raises the broader open question of whether the emerging privacy-heterogeneity cost tradeoff is best balanced by model personalization, as many existing methods including MR-MTL inherently require at least one hyperparameter to specify "how much to personalize" for general utility improvements over local training and FedAvg. Alternatively, the hyperparameter(s) may be estimated *during* training (approach (3) in the first paragraph), though such procedures may not be general and/or scalable and may need to be tailored to the specific personalization method. On the positive side, it is unclear whether the choice of $\lambda$ can meaningfully leak privacy in practice. MR-MTL may also be viewed favorably as a strong baseline since it only needs one hyperparameter to attain its benefits, while other existing methods that require more tuning will incur even larger privacy costs from hyperparameter tuning.

## 8 Concluding Remarks

In this work, we revisit the application of differential privacy in cross-silo FL. We examine silo-specific sample-level DP as a more appropriate privacy notion for cross-silo FL, and we point out several meaningful ways in which it differs from client-level DP commonly studied under the cross-device setting, particularly when analyzing tensions between privacy, utility, and heterogeneity. We explore and establish baselines under this privacy setting and identify desirable properties for a personalization method for balancing an emerging tradeoff between utility costs from privacy and heterogeneity. We then analyze a simple, promising method (MR-MTL) and discuss key open questions for the area at large. Some future directions include (1) extending the privacy model to cases where data subjects have multiple records across silos, (2) extending our theoretical characterization to deep learning cases or performing a large-scale empirical study, and (3) developing auto-tuning algorithms for personalization hyperparameters with minimal privacy overhead.

**Acknowledgements.** We thank Sebastian Caldas, Tian Li, Yash Savani, Amrith Setlur, and Peter Kairouz for helpful discussions and feedback and Thomas Steinke for guidance on implementing privacy accounting for hyperparameter tuning [79]. This work was supported in part by the NSF Grants IIS1838017 and IIS2145670, a Meta Faculty Award, an Apple Faculty Award, the Intel Private AI Center, and the CONIX Research Center. ZSW was supported in part by the NSF Award CNS2120667. Any opinions, findings, and conclusions or recommendations expressed herein are those of the author(s) and do not necessarily reflect the NSF or any other funding agency.

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
