# Appendix

# A  Additional Background

**Rényi Differential Privacy (RDP).**  In this work, we make use of a relaxation of different privacy known as Rényi Differential Privacy [77] for tight privacy accounting.

**Definition A.1** (Rényi Differential Privacy (RDP) [77])**.** A randomized algorithm $M : \mathcal{X}^n \to \mathcal{Y}$ is $(\alpha, \varepsilon)$-RDP with order $\alpha > 1$ if for any adjacent datasets $x, x' \in \mathcal{X}^n$,

$$D_\alpha(M(x) \| M(x')) \le \varepsilon, \tag{11}$$

where $D_\alpha(P\|Q)$ is the Rényi divergence[6] between distributions $P$ and $Q$:

$$D_\alpha(P\|Q) \triangleq \frac{1}{\alpha - 1} \log \mathop{\mathbb{E}}_{x \sim P} \left[ \left( \frac{P(x)}{Q(x)} \right)^{\alpha - 1} \right]. \tag{12}$$

Under Rényi DP, the privacy composition is simple: if every step of an algorithm satisfies $(\alpha, \varepsilon)$-RDP, then over $T$ steps the algorithm satisfies $(\alpha, T\varepsilon)$-RDP. The following lemma from [15, 17] provides a conversion from RDP to standard $(\varepsilon, \delta)$-DP guarantees.

**Lemma A.2** (Conversion from Rényi DP to approximate DP [15, 17])**.** *If a mechanism $M$ satisfies $(\alpha, \varepsilon(\alpha))$-RDP, then for any $\delta > 0$, it also satisfies $(\varepsilon(\delta), \delta)$-DP where*

$$\varepsilon(\delta) = \inf_{\alpha > 1} \varepsilon(\alpha) + \frac{1}{\alpha - 1} \log \left( \frac{1}{\alpha \delta} \right) + \log \left( 1 - \frac{1}{\alpha} \right). \tag{13}$$

**Zero-Concentrated Differential Privacy (zCDP).** A closely related privacy notion is zero-concentrated DP (zCDP [15]), where $\rho$-zCDP is equivalent to satisfying $(\alpha, \rho\alpha)$-Rényi DP simultaneously for all orders $\alpha$. Thus, algorithms that satisfy zCDP guarantees are compatible with standard RDP accounting routines implemented in open-source libraries (e.g. TensorFlow Privacy [72]). In our work, we make use of zCDP and a related result for the Exponential Mechanism [87] for tight privacy composition when implementing private cluster selection for IFCA [33, 69] (see below and Appendix C.3).

**Exponential Mechanism for Private Selection.** The Exponential Mechanism (EM) is a standard algorithm for making private selection from a set of candidates based on their scores [75]. Specifically, there is a dataset $x \in \mathcal{X}^n$ requiring DP protection, and a scoring function $s : \mathcal{X}^n \times [G] \to \mathbb{R}$ that evaluates a set of candidates $g \in [G]$. We want to pick the candidate with the highest score (i.e. $\operatorname{argmax}_{g \in [G]} s(x, g)$) subject to $(\varepsilon, 0)$-DP for neighboring datasets $x, x'$. The mechanism $M$ is defined by setting the probability of choosing any $g \in [G]$ as

$$\Pr[M(x) = g] = \frac{\exp \left( \frac{\varepsilon}{2\Delta} \cdot s(x, g) \right)}{\sum_{g' \in [G]} \exp \left( \frac{\varepsilon}{2\Delta} \cdot s(x, g') \right)}, \tag{14}$$

where $\Delta$ is the sensitivity of the scoring function. EM also satisfies $\frac{1}{8}\varepsilon^2$-zCDP [87] and thus $(\alpha, \frac{\alpha}{8}\varepsilon^2)$-RDP for all $\alpha$. A variant of EM is the Permute-and-Flip mechanism [70].

The Exponential Mechanism can be implemented as "Report Noisy Max" with Gumbel noise: we can add independent noises drawn from the Gumbel distribution with scale $\frac{2\Delta}{\varepsilon}$ to the candidate scores $s(x, g)$ for all $g \in [G]$ and simply report the max noisy score. If the score function is a loss metric (where we want the minimum instead of the maximum), we can similarly implement "Report Noisy Min" by subtracting the Gumbel noises from the scores and report the minimum. The latter is used in our implementation for private cluster selection, where clients select clusters with lowest loss or error rate (i.e. $1 - \text{accuracy}$); see Appendix C.3 for more details.

**Privacy Budgeting.** A typical accounting workflow, as used in our experiments, thus involves (1) composing the RDP guarantees of all private operations in the algorithm and (2) trying a list of $\alpha$ values that give the lowest $\varepsilon$ for a target $\delta$ when converting back to $(\varepsilon, \delta)$-DP that captures the overall privacy cost. For SGD training, we also use existing results on privacy amplification via subsampling [78, 1]: if a gradient step is $(\varepsilon, \delta)$ w.r.t. the dataset *without* amplification and the gradient is computed with a minibatch (assumed to be a random sample) of batch size $b = q/n$ where $q$ is the sampling ratio and $n$ is the size of the dataset, then the privacy of the gradient step is amplified to $(O(q\varepsilon), \delta)$-DP. In a silo-specific sample-level DP setup, the size of the dataset $n_k$ at each silo thus

---

[6]The Rényi divergence at $\alpha = 1$ is defined as $D_1(P\|Q) \triangleq \mathop{\mathbb{E}}_{x \sim P} \left[ \log \left( \frac{P(x)}{Q(x)} \right) \right] = \lim_{\alpha \to 1} D_\alpha(P\|Q)$, which is also the KL divergence.

determines the extent of the amplification, and thus even if silos target for the same $(\varepsilon, \delta)$ sample-level DP, they may end up adding different amounts of noise when running DP-SGD (mentioned in the **Training Setup** paragraph of §4). For this reason our experiments primarily focus on having the same privacy target for all silos.

**Heterogeneous Differential Privacy.** A related privacy notion is *heterogeneous DP* [5, 47], where each item within a dataset to be protected by DP may opt for a different $(\varepsilon, \delta)$ target. Our setting primarily focuses on different $(\varepsilon, \delta)$ values for disjoint datasets, and all items within a specific dataset share the same DP target.

See also Appendix F for additional background relating to Section §7 (private hyperparameter tuning).

## B  Additional Discussions

### B.1  Limitations

We discuss below the limitations of this work in addition to Section §7.

**When multiple records map to the same entity.** In this paper we studied the application of silo-specific sample-level differential privacy in cross-silo federated learning. While this is an important initial step towards a more suitable privacy model for cross-silo FL (in contrast to the commonly studied client-level DP model), we assume that each entity that requires privacy protection has at most one record (training example) across silos (e.g. a single patient has one medical record at a hospital).

There are two characteristic cases where this assumption does not hold for all items in a silo:

- **Multiple records within a silo map to the same entity.** One example would be a student re-enrolling at the same school for multiple degree programs, thus creating multiple student records at the same silo. In such cases, the silo curator may need to carefully apply group privacy or other methods for ensuring entity-level privacy [55] to protect the entity rather than its records.

- **Multiple records across silos map to the same entity.** One example would be a person having multiple credit cards at different banks. This case is more intricate as it is harder to precisely account for the DP guarantee for this entity without knowing (1) the silos in which this entity has appeared and (2) the specific privacy targets for each of those silos. In this case, the silos may cooperate to run *private set intersection* (e.g. as considered in [67]) to privately identify this scenario, but this would by itself come at a privacy cost.

These cases are interesting avenues for future research on private cross-silo learning.[7]

**Extending the analysis to deep learning cases.** In Section §6 we use federated mean estimation as a simplified setting for analyzing the behavior of MR-MTL under silo-specific sample-level privacy. While the analysis provides adequate insights into the empirical phenomena in Fig. 5, it is a simple model that does not consider the dynamic aspects of the learning settings, including (1) the Gaussian random walk component of the model updates due to DP noise applied over many training rounds, (2) the effect of communication frequency on the effect of noise reduction, (3) the concept of "client drifts" (as considered in [52]) as a result of heterogeneity and how it interfaces with the DP noises, and (4) how overparameterization may affect all of the above.

**Caveats of cross-silo learning with very large local datasets.** In contrast to cross-device federated learning, cross-silo federated learning is typically characterized by having a limited number of clients, each with a large local dataset. The term "large" is relative because it describes the sufficiency of the datasets for fitting good local models of a *specific class*; for example, 500 examples are likely sufficient to fit linear regression of 10 parameters, but very likely insufficient to learn a transformer [98]. In this sense, many FL problems in practice – such as large commercial banks running regression on tabular data – will in fact have local training to be the *optimal* strategy, as long as there are sufficient local data and the data from other silos are not of the same local distribution. In these cases, one should expect MR-MTL to opt for $\lambda^* \approx 0$ as federated learning is not needed at all, and thus its advantages under privacy will also be minimal.

---

[7]The case of having individual records corresponding to multiple entities at once (e.g. one record for all family members) is slightly less interesting since sample-level privacy would protect all of the entities.

## B.2 Potential Negative Societal Impact

Our work studies the empirical behaviors that arise when applying an alternative model of differential privacy to cross-silo federated learning, and we provide a strong baseline method (MR-MTL) that fares well in this setting. In this sense, our work sheds light on and facilitates the development of a previously underexplored area of differentially private federated learning. However, because MR-MTL requires selecting a good regularization stength $\lambda$, one potential negative impact is that users may excessively tune $\lambda$ on a private dataset and inadvertently leak privacy via the choice of $\lambda$ (perhaps qualitatively rather than quantitatively); for example, if a silo chose a large $\lambda$ for better performance, then in principle its data would look somewhat more similar to the "average" of the silo datasets. Moreover, our privacy model requires that silos add their own independent noises for their own DP targets, and this requirement may not be followed correctly (either deliberately or inadvertently) to provide vacuous DP guarantees for people's data.

## C Additional Experimental Details

### C.1 Datasets and Models

Table A1 summarizes the datasets, tasks, and models considered in our experiments. In the following, we provide details on each.

| Dataset | Task | # Clients (Silos) | Input Dim | Min $n_k$ | Max $n_k$ | Learner |
|---|---|---|---|---|---|---|
| Vehicle | Classification | 23 | 100 | 872 | 1933 | SVM |
| School | Regression | 139 | 28 | 15 | 175 | Linear |
| Google Glass (GLEAM) | Classification | 38 | 180 | 699 | 776 | SVM |
| Heterogenous CIFAR-10 | Classification | 30 | $32 \times 32 \times 3$ | 1515 | 1839 | ConvNet |
| Rotated & Masked MNIST | Classification | 40 | $28 \times 28 \times 1$ | 1500 | 1500 | ConvNet |
| Subsampled ADNI | Regression | 9 | $32 \times 32 \times 1$ | 45 | 2685 | ConvNet |

Table A1: Summary of datasets, tasks, and models for our empirical studies. $n_k$ denotes the number of training examples on client $k$.

**Vehicle [26].** The Vehicle Sensor dataset is a binary classification dataset containing $K = 23$ data silos. Each silo (sensor) has acoustic and seismic measurements for a road segment, with each data point being a 100-dimensional feature vector describing the measurements when a vehicle passes through the road segment. The goal is to predict between two predetermined types of vehicles. We use a train/test split of 75%/25% following previous work [91], yielding $872 \leq n_k^{\text{train}} \leq 1933$ training examples on each client. We use simple linear SVMs for classification following [91, 57]. It is a suitable dataset for cross-silo FL because the number of silos $K$ is small while each silo has sufficient data to fit a good local model, as opposed to cross-device datasets such as FEMNIST [16] where $K$ is large but each silo has little data to learn a useful model. Moreover, we can use tight privacy budgets due to reasonably large local datasets (in terms of sufficiency for fitting a good local model) and SVMs (which are relatively noise-tolerant since decision boundaries only depend on support vectors). The dataset is accessible from the original authors.[8]

**School [35, 9, 111].** The School dataset originated from the now-defunct Inner London Education Authority.[9] It is a regression dataset for predicting the exam scores of 15,362 students distributed across 139 secondary schools. Each school has records for between 22 and 251 students, and each student is described by a 28-dimensional feature vector capturing attributions such as the school ranking, student birth year, and whether the school provided free meals. We perform 80%/20% train/test split in Fig. 3 (with $15 \leq n_k^{\text{train}} \leq 175$ training examples in each silo), and additionally consider 50%/50% and 20%/80% train/test split in Fig. A8. We use simple linear regression models following previous work (e.g. [9, 111, 36]) to predict student scores. Like the Vehicle dataset, the School dataset is a natural cross-silo FL dataset with a limited number of clients $K$, each with roughly sufficient data to fit a reasonable local model. The dataset is available from [111].

---

[8]https://web.archive.org/web/20110515133717/http://www.ece.wisc.edu:80/~sensit/
[9]https://en.wikipedia.org/wiki/Inner_London_Education_Authority

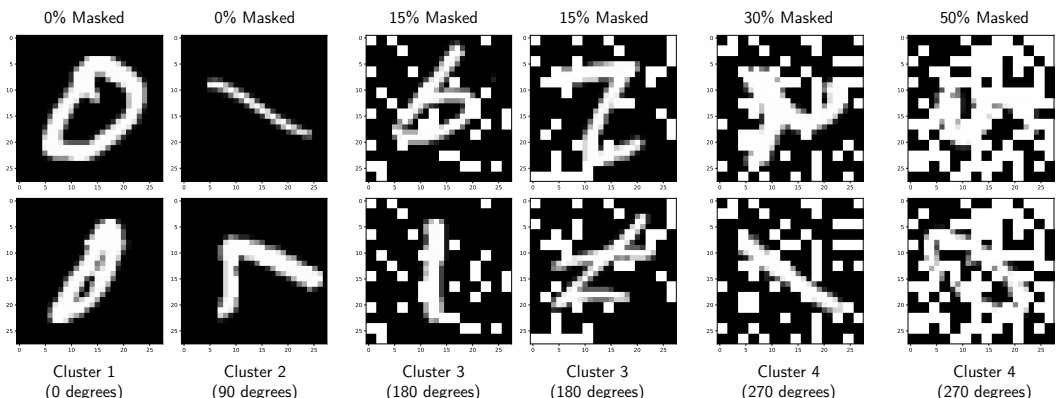

Figure A1: **Example images of the Rotated & Masked MNIST dataset**. Each column corresponds to two images from the same (random) client from the specified cluster (thus they have the same client-specific random mask). Labels for each column from left to right: (0, 0), (1, 7), (9, 1), (2, 7), (2, 1), (5, 0).

**Google Glass Eating and Motion (GLEAM) [82].** We also benchmark on the GLEAM dataset, a real-world head motion tracking dataset for binary classification. The motion data is collected with Google Glass, and the task is to classify the activity of the wearer (eating or not). There are in total $K = 38$ silos (wearers) and 27800 data points, with each silo containing $699 \leq n_k \leq 776$ data points. Each data point is a 180-dimensional feature vector capturing head movement of the wearers. Linear models yield reasonable utility on GLEAM and thus we use linear SVMs following previous work [91]. Like Vehicle and School, this is a suitable dataset for cross-silo FL given a small $K$ and relatively large local datasets.

**Heterogeneous CIFAR-10 Dataset.** We additionally evaluate on CIFAR-10 [53], with heterogeneous client data split following previous work [94, 90] (based on the code provided by [90]). The dataset has $K = 30$ clients (silos) in total, and data heterogeneity is generated with each client having a random number of samples from 5 randomly chosen classes out of the 10 classes. Each client has $1515 \leq n_k \leq 1839$ training examples. We use a simple convolutional network with the following layers: [Conv $3 \times 3$ with 32 channels, ReLU, MaxPool $2 \times 2$ with stride 2, Conv $3 \times 3$ with 64 channels, ReLU, MaxPool $2 \times 2$ stride 2, Linear]. No padding is used for convolutional layers.

**Rotated & Masked MNIST.** We adapt the original MNIST dataset [54] to study the effect of structured heterogeneity on MR-MTL. For the 60000/10000 train/test images, we first perform a shuffle and then evenly separate them into $K = 40$ clients (silos), each with 1500/250 train/test images with roughly uniform distribution on the labels. We then randomly separate the silos into 4 groups of 10, and apply rotations of $\{0°, 90°, 180°, 270°\}$ to each group respectively; silos within the same group have the same rotations applied to the images, thus forming 4 natural silo clusters. To add *intra-cluster* heterogeneity, we then apply *silo-specific* random masks of $2 \times 2$ white patches; that is, all images in the same silo has the same mask, and no two silos have the same mask with very high probability. The random masks are akin to those considered in [40]. The white patches of the random mask do not overlap, and the mask ratio is the probability of a patch being applied (so the specified percentage of masked area is an expectation). Examples of generated images are shown in Fig. A1. These image transformations introduce two types of heterogeneity identified by [50]: "covariate shift" (skew of feature distributions) and "concept drift" (same label, different features). The model architecture is the same as the one used for heterogeneous CIFAR-10.

**Subsampled Alzheimer's Disease Neuroimaging Initiative (ADNI) Dataset [2].** We additionally benchmark on the ADNI dataset, which is a real-world dataset containing brain PET scans of Alzheimer's disease patients, patients with mild cognitive impairment, and healthy people taken from multiple institutions [81]. It is a regression dataset for predicting the SUVR value (a scalar ranged roughly between 0.8 and 2) from the PET scan images of a brain. We simplified the full dataset for faster training by subsampling the axial slices generated for each brain PET scan (96 slices for scan), turning them into a gray scale image, downsampling them to size $32 \times 32$, and randomly splitting them into a 75%/25% train/test sets. There are in total $K = 9$ silos containing a total of 11040 images; each silo corresponds to a different equipment that took the PET scans and contains $45 \leq n_k^{\text{train}} \leq 2685$ training examples. See Fig. 4 of [86] and Fig. 4 of [81] for sample images. The

model architecture is a simple convolutional network with the following layers: [Conv $5 \times 5$ with 32 channels, ReLU, MaxPool $2 \times 2$ with stride 2, Conv $5 \times 5$ with 64 channels, ReLU, MaxPool $2 \times 2$ stride 2, Linear]. No padding is used for convolutional layers.

### C.2 License/Usage Information for Datasets

**Vehicle.** The Vehicle dataset was made publicly available by the original authors as a research dataset [26] and license information was unavailable. It has been subsequently used in many work (e.g. [91]).

**School.** The original entity that collected the School dataset [35] is defunct and license information was unavailable. The dataset has been made publicly available [111] and used extensively in previous work (e.g. [111, 107]).

**Google Glass (GLEAM).** The GLEAM dataset was made publicly available by the original authors and can be used for any non-commercial purposes. See this URL[10] for license and usage information.

**Heterogeneous CIFAR-10.** The original CIFAR-10 dataset is available under the MIT license.

**Rotated & Masked MNIST.** The original MNIST dataset is available under the CC BY-SA 3.0 license.

**Subsampled ADNI.** As per the Data Use Agreement of the ADNI dataset:[11]

> Data used in preparation of this manuscript were obtained from the Alzheimer's Disease Neuroimaging Initiative (ADNI) database (adni.loni.usc.edu). As such, the investigators within the ADNI contributed to the design and implementation of ADNI and/or provided data but did not participate in analysis or writing of this manuscript. A complete listing of ADNI investigators can be found at this URL.[12]

The data sharing and publication policy of the ADNI dataset can be found at this URL.[13] Access to the dataset must be approved by ADNI.

### C.3 Benchmark Methods and Implementation

We provide more details on our benchmark personalization methods below.

**Local finetuning** [99, 104, 19] is one of the simplest but most effective personalization methods: once clients obtain a shared model via federated training (e.g. FedAvg), they can personalize it with additional training steps over their local dataset. This simple strategy has been shown to work very well empirically [99, 104, 19], with the work of [19] providing theoretical support that it can asymptotically achieve comparable performance to other more sophisticated methods. Moreover, in contrast to other local adaptation methods like distillation [104], local finetuning's privacy footprint under our privacy model can be easily controlled (by limiting the number of finetuning and total training steps) without qualitatives change in its behavior. In our experiments, local finetuning is implemented as FedAvg followed by local training, each taking 50% of the total number of training rounds to ensure an identical privacy budget as other baseline methods.

**Ditto** [57] is the current state-of-the-art method for personalization with provable benefits of robustness and fairness. It is closely related to MR-MTL because it similarly trains personalized models while $\ell^2$-regularizing them towards a global model, but it differs from MR-MTL in that its global model can be obtained by a standalone solver. In particular, when no privacy is added, Ditto's modularity allows it to perfectly interpolate between local and FedAvg training with its regularization strength $\lambda$. In our experiments, we implement Ditto with the FedAvg solver and use a minimal number of iterations over the local datasets to avoid excessive privacy overhead.

---

[10]http://www.healthailab.org/data.html
[11]https://adni.loni.usc.edu/wp-content/uploads/how_to_apply/ADNI_Data_Use_Agreement.pdf
[12]https://adni.loni.usc.edu/wp-content/uploads/how_to_apply/ADNI_Acknowledgement_List.pdf
[13]https://adni.loni.usc.edu/wp-content/uploads/how_to_apply/ADNI_DSP_Policy.pdf

**Mocha** [91] is a multi-task learning framework tailored for federated settings. During training, it simultaneously learns the personalized models as well as a client-relationship matrix which can model both positive and negative client relationships (in contrast, clustering methods only focus on positive relationships). One disadvantage of Mocha is that it applies to convex problems only. In particular, the original paper uses a dual formulation for efficient training, but for fair comparison and compatibility with privacy primitives (especially DP-SGD [92, 11, 1]), we implemented Mocha in its primal form and trained it with SGD in our experiments. Similarly, we align Mocha to other baselines in terms of the number of training steps to prevent privacy overhead.

**IFCA** [33] (and the conceptually similar **HypCluster** [69]) is a simple clustering framework proposed as an extension to FedAvg. In every round of IFCA training, the server sends $k$ models (cluster centroids) to all clients; each client locally evaluates them over its local training data and selects the one with the lowest loss. Each client then locally trains on the selected model and returns updates only for this model, along with its index. Clients that selected the same model indices can be viewed as belonging to the same cluster. We use IFCA as the representative clustering method due to its performance, simplicity, and practicality.

The privacy overhead of IFCA comes in the form of *private cluster selection*: when each client evaluates the incoming models (cluster centroids) on the local datasets, this procedure must be privatized as the selection itself may leak information about the dataset. Private selection can be implemented via the exponential mechanism [75] with sharp accounting via a bounded range analysis [87] (discussed in Appendix A), but one must decide (e.g. as a privacy budget $\varepsilon_{\text{select}}$ for the same $\delta$) how to share the selection cost with DP-SGD under a fixed total privacy budget $\varepsilon_{\text{total}}$.

**Mitigating strategies for private cluster selection.** In our experiments, we observed that if private selection is implemented naively, it can incur a prohibitive privacy overhead and destroys the final utility (e.g. Fig. 4). There are two important reasons: (1) unlike DP-SGD, no privacy amplification applies to private selection, and (2) the sensitivity of the training loss (which is used to select cluster centroids) is unbounded in general and must be clipped to a reasonable value (e.g. $\leq 1$). We propose two mitigation strategies:

1. **Use accuracy instead of loss.** For the cluster selection metric (i.e. the score function $s(x, g)$ where $x$ is the local dataset and $g$ is a particular cluster centroid), we use the error rate ($1 -$ accuracy) instead of the loss (which is used by the original authors [33, 69]). The rationale is that accuracy is a low-sensitivity function, particularly in cross-silo settings: one can show that, by enumerating the cases where the differing example between the neighboring datasets $x$ and $x'$ are correctly/incorrectly classified under addition/removal/replacement notions of DP, the sensitivity $\Delta_{\text{acc}}$ of $s$ is bounded as

$$\Delta_{\text{acc}} = \max_g \max_{x,x'} |s(x, g) - s(x, g')| \leq \frac{1}{n-1} \tag{15}$$

   where $n$ is the size of the local dataset. Since $n$ can be large in cross-silo settings, the sensitivity can be orders of magnitude smaller than that of the loss function. With small sensitivity, we heuristically set the per-round selection privacy budget to a very small $\varepsilon_{\text{select}} = 0.03 \cdot \varepsilon_{\text{total}}$. Despite this, however, the cost of private selection can still grow quickly over the entire training process and considerably eat into DP-SGD's privacy budget.

2. **Truncate the number of cluster selection rounds.** Our second strategy is to simply run less rounds of cluster selection (e.g. to 10% of total number of training rounds, as in Fig. 3). This follows from the empirical analysis of [33] as well as our own experimental observation that clusters tend to converge quickly, though in some cases, clusters may not fully converge within 10% of training rounds.

Despite these strategies, however, the private selection cost can still lead to a steep utility hit. Note also that the new hyperparameter $\varepsilon_{\text{select}}$ can be tuned; for a fixed total budget $\varepsilon_{\text{total}}$, a small $\varepsilon_{\text{select}}$ means the budget for DP-SGD $\varepsilon_{\text{train}}$ is not affected by much, but the selected clusters would be very noisy and inaccurate; a large $\varepsilon_{\text{select}}$ leads to less noisy clusters (and thus smaller intra-cluster heterogeneity), but DP-SGD will correspondingly use larger noise and hurt optimization.

### C.4   Training Settings

**Optimizers.** For simplicity of hyperparameter tuning and experimental controls, we use minibatch DP-SGD for client local training without local or server momentum for all experiments (in fact,

FedAvgM [43] and FedAdam [85] were not found to be helpful on Vehicle and School). While there are more efficient solvers for the Vehicle and School datasets since we are dealing with convex problems, we want compatibility with DP-SGD [92, 11, 1] as well as tight privacy accounting with privacy amplification via subsampling (i.e. via minibatch training).

**Hyperparameters.** For all datasets and all methods, we set silos to train for 1 local epoch in every round (except Ditto [57] which takes 2 local epochs). For Vehicle, GLEAM, School, Heterogeneous CIFAR-10, Rotated & Masked MNIST, and subsampled ADNI respectively, the local batch size across all silos are fixed with $B = 64, 64, 32, 100, 100, 64$, and the clipping norm for per-example gradients are heuristically set to $c = 6, 6, 1, 8, 1, 0.5$. Vehicle uses $T = 400$ rounds for most experiments (except Fig. 2 which trains for $T = 200$ rounds); School, Google Glass, Heterogeneous CIFAR-10, and Rotated & Masked MNIST use $T = 200$; and ADNI uses $T = 500$.

For multi-task learning methods (MR-MTL, Ditto [57], Mocha [91]), we sweep the regularization strength across a grid of $\lambda \in [0.0001, 0.001, 0.003, 0.01, 0.03, 0.1, 0.3, 1, 3, 10]$ to find the best $\lambda^*$ wherever applicable (e.g. Figs. 3, 5 and A8). To compensate for the change in the gradient magnitude, we also sweep different client learning rates across a grid of $\eta \in [0.001, 0.003, 0.01, 0.03, 0.1, 0.3]$; for fair comparison, the same grid of $\eta$ is swept for methods that do not involve $\lambda$ (e.g. IFCA, local finetuning).[14] For Fig. 2 and Fig. A2, the learning rate is fixed to $\eta = 0.01$. For all datasets, the chosen privacy parameter $\delta$ satisfy $\delta < n_k^{-1.1}$ where $n_k$ is the local training dataset size.

**Evaluation Protocol.** For all datasets, we evaluate methods by the average test metric (accuracy or MSE) across the silos, weighted by their respective test sample counts. Weighted averaging allows the final test metric to reflect a method's performance over the individual test samples of the combined dataset across silos, thus fairer and more aligned (compared to uniform averaging of silo test metrics) with our privacy model where each test sample represents an entity requiring protection.

## C.5 Hardware

Experiments for Vehicle, School, and Google Glass (GLEAM) are run on commodity CPUs and experiments for Heterogeneous CIFAR-10, Rotated & Masked MNIST and ADNI are run on four NVIDIA RTX A6000 GPUs.

## C.6 Code

Our experiments are implemented in Python with NumPy [39], JAX [14] and Haiku [42]. For private training, JAX is used to vectorize DP-SGD over per-example gradients [93]. Code is available at https://github.com/kenziyuliu/private-cross-silo-fl.

# D  Additional Algorithmic Details

## D.1  Mean-Regularized Multi-Task Learning (MR-MTL)

Algorithm A1 describes the canonical instantiation of MR-MTL [94]. Its key ingredient is the mean-regularization (Line 6 of Algorithm A1) that forces the local personalized models $w_k$ to be close to their mean $\bar{w}$. Silo-specific sample-level privacy is added by privatizing the local gradients as in DP-SGD [92, 11, 1].

**Privacy of MR-MTL.** Since the iterates of $\bar{w}^{(t)}$ are already differentially private (as they are the average of the private iterates $w_k^{(t)}$), the additional regularization term $\frac{\lambda}{2}\|w_k^{(t)} - \bar{w}_k^{(t-1)}\|_2^2$ (and hence MR-MTL) does not incur privacy overhead compared to local and FedAvg training.

**Weighted vs Unweighted Model Updates.** It is customary to weigh the model updates from each client (silo) by its training example counts, as in the original FedAvg implementation [71]. However, note that under silo-specific sample-level privacy with *addition/removal* notions of DP, the example

---

[14]As discussed in Section §7, releasing the results from repeated experiments (possibly with different hyperparameters) may technically compromise the privacy of the datasets [62, 79]. In our case, we are primarily interested in understanding the behaviors and tradeoffs that emerge under silo-specific sample-level DP, and thus for experimental control and ease of comparison we do not account for the privacy costs from hyperparameter tuning and repetitions. We also use only public datasets in our experiments.

---

**Algorithm A1** Mean-Regularized Multi-Task Learning

---

1: **Input:** Initial client models $\{w_k^{(0)}\}_{k\in[K]}$, and mean model $\bar{w}^{(0)}$.
2: **for** training round $t = 1, ..., T$ **do**
3:     Server sends $\bar{w}^{(t-1)}$ to every client.
4:     **for** client $k = 1, ..., K$ **in parallel do**
5:         Set model iterate: $w_k^{(t)} \leftarrow w_k^{(t-1)}$.
6:         For every batch $(x, y)$, client updates $w_k^{(t)}$ using SGD or DP-SGD with batch loss
         $\ell(w_k^{(t)}, x, y)$ and gradient
$$\nabla_{w_k^{(t)}} \left[ \ell\left(w_k^{(t)}, x, y\right) + \frac{\lambda}{2} \left\| w_k^{(t)} - \bar{w}^{(t-1)} \right\|_2^2 \right].$$
7:         Return model update $\Delta_k^{(t)} = w_k^{(t)} - w_k^{(t-1)}$.
8:     Server updates $\bar{w}^{(t)} = \bar{w}^{(t-1)} + \frac{1}{K} \sum_{k=1}^{K} \Delta_k^{(t)}$ (may weigh $\Delta_k^{(t)}$ by client example counts).
9: **Output:** Personalized models $w_k^{(T)}$ for all $i \in [K]$.

---

counts on each silo may itself leak sensitive information (e.g. when a silo only has one record). This is less of an issue with the *replacement* notion of DP, since neighboring datasets would have the same example counts. In our experiments we use weighted model updates and thus implicitly assume the replacement notion of DP. Nevertheless, the resulting privacy guarantees are constant factors apart and empirically we did not observe significant changes in performance when using unweighted aggregation.

### D.2   IFCA Preconditioning / Warm-Starting

In Section §5, we considered an extension to MR-MTL where training is "warm-started" by a small number of rounds of clustering (via IFCA [33, 69]), followed by MR-MTL training *within* each formed cluster. Pseudocode for this procedure is shown in Algorithm A2.

**Privacy of IFCA Preconditioning.** Observe that as with MR-MTL, the gradient steps satisfy silo-specific sample-level DP regardless or whether the steps are made on the cluster models or the personalized models. IFCA preconditioning introduces privacy overhead in the form of private cluster selection (Line 6 of Algorithm A2; discussed in Appendices A and C.3), which splits the total privacy budget with DP-SGD. As a result the noise scale for DP-SGD would be increased and can be numerically determined.

---

**Algorithm A2** IFCA-Preconditioned MR-MTL

---

1: **Input:** Initial client models $\{w_k^{(0)}\}_{k\in[K]}$, number of clusters $G$, initial cluster models $\{\bar{w}_g^{(0)}\}_{g\in[G]}$, total number of rounds $T$, and the number of initial clustering rounds $T_{\text{cluster}}$.

2: `# IFCA preconditioning rounds`

3: **for** IFCA training round $t = 1, ..., T_{\text{cluster}}$ **do**

4:      Server sends cluster models $\{\bar{w}_g^{(t-1)}\}_{g\in[G]}$ to every client.

5:      **for** client $k = 1, ..., K$ **in parallel do**

6:          **Use Exponential Mechanism (Appendix A) to select best cluster** $\bar{w}_{g^*(k)}^{(t-1)}$ from $\{\bar{w}_g^{(t-1)}\}$ with loss/error rate function $s$ and local dataset $(X_k, Y_k)$.

7:          Set model iterate: $w_k^{(t)} \leftarrow \bar{w}_{g^*(k)}^{(t-1)}$.

8:          For every batch $(x, y)$, client updates $w_k^{(t)}$ using SGD or DP-SGD with batch loss $\ell(w_k^{(t)}, x, y)$ and gradient $\nabla_{w_k^{(t)}} \ell\left(w_k^{(t)}, x, y\right)$.

9:          Return model update $\Delta_k^{(t)} = w_k^{(t)} - \bar{w}_{g^*(k)}^{(t-1)}$ and selected cluster index $g^*(k)$.

10:      **for** each cluster $g \in [G]$ **do**

11:          Server applies (weighted) model updates to cluster $g$ with the associated client indices:

$$\bar{w}_g^{(t)} = \bar{w}_g^{(t-1)} + \frac{1}{|\mathcal{K}_g|} \sum_{k\in\mathcal{K}_g} \Delta_k^{(t)}, \quad \text{where } \mathcal{K}_g \equiv \{k \in [K] \mid g^*(k) = g\}. \tag{16}$$

12: `# MR-MTL rounds with regularization towards frozen cluster centroids`

13: **for** MR-MTL training round $t = T_{\text{cluster}} + 1, ..., T$ **do**

14:      Server sends cluster models $\{\bar{w}_g^{(t-1)}\}_{g\in[G]}$ to every client.

15:      **for** client $k = 1, ..., K$ **in parallel do**

16:          **Client retrieves the last selected cluster centroid** $\bar{w}_{g^*(k)}^{(t-1)}$.

17:          Set model iterate: $w_k^{(t)} \leftarrow w_k^{(t-1)}$ **(using the personalized model from last round)**.

18:          For every batch $(x, y)$, client updates $w_k^{(t)}$ using SGD or DP-SGD [92, 11, 1] with batch loss $\ell(w_k^{(t)}, x, y)$ and gradient

$$\nabla_{w_k^{(t)}} \left[ \ell\left(w_k^{(t)}, x, y\right) + \frac{\lambda}{2} \left\| w_k^{(t)} - \bar{w}_{g^*(k)}^{(t-1)} \right\|_2^2 \right].$$

19:          Return model update $\Delta_k^{(t)} = w_k^{(t)} - \bar{w}_{g^*(k)}^{(t-1)}$ and the cluster index $g^*$.

20:      **for** each cluster $g \in [G]$ **do**

21:          Server applies (weighted) model updates to cluster $g$ with the associated client indices:

$$\bar{w}_g^{(t)} = \bar{w}_g^{(t-1)} + \frac{1}{|\mathcal{K}_g|} \sum_{k\in\mathcal{K}_g} \Delta_k^{(t)}, \quad \text{where } \mathcal{K}_g \equiv \{k \in [K] \mid g^*(k) = g\}. \tag{17}$$

22: **Output:** Personalized models $w_k^{(T)}$ for all $i \in [K]$.

---

# E  Additional Analysis Details

In this section we provide additional details for the analysis presented in Section §6. We also extend the analysis to the case with varying $n$, $\sigma$, $\sigma_{\mathrm{DP}}$ for each silo in Appendix E.5.

## E.1  Notations

Unless otherwise specified, we used the following notations throughout the analysis:

- $K$ denotes the total number of silos (clients) with indices $k \in [K]$;
- $n$ denotes the number of examples on each silo;
- $w_k$ denotes the true center of silo $k$'s data distribution;
- $X_k \equiv \{x_{k,i}\}_{i \in [n]}$ denotes the local dataset with $n$ data points;
- $\hat{w}_k$ denotes the best local estimator of $w_k$;
- $\bar{w}$ denotes the average of local estimators;
- $\hat{w}_{\setminus k} \triangleq \frac{1}{K-1} \sum_{j \neq k, j \in [K]} \hat{w}_j$ denotes the external average estimators from the perspective of $k$;
- $\sigma^2$ denotes the sampling variance of the local data $X_k$;
- $\tau^2$ denotes the sampling variance of the local data centers $w_k$ (hence a measure of data heterogeneity across silos); and
- $\sigma_{\mathrm{DP}}^2$ denotes the DP noise variance on each silo to satisfy silo-specific sample-level privacy.

## E.2  MR-MTL Formulation

The general formulation of the mean-regularized MTL objective may be expressed as

$$\min_{w_k, k \in [K]} \frac{1}{K} \sum_{k=1}^{K} h_k(w_k) \quad \text{with} \quad h_k(w) \triangleq F_k(w) + \frac{\lambda}{2} \|w - \bar{w}\|_2^2, \tag{18}$$

where $F_k(\cdot)$ is the local objective for client $k$, $\bar{w} \triangleq \frac{1}{K} \sum_{k=1}^{K} w_k$ is the average model, and $\lambda \geq 0$ is the regularization strength. A larger $\lambda$ enforces the models to be closer to each other, and $\lambda = 0$ reduces the problem to local training. In particular, unlike Ditto [57], MR-MTL may *not* recover FedAvg [71] under SGD as $\lambda \to \infty$, since $\lambda$ essentially changes the ratio between the local objective gradients and the regularization gradients. For the purposes of our analysis, we assume $F_k$ is strongly convex for all $k \in [K]$.

## E.3  Assumptions

Our characterization of MR-MTL on makes the following simplifying assumptions.

**Assumption E.1.** All clients (silos) have the same number of data points $n$, data sampling variance $\sigma^2$, and DP noise variance $\sigma_{\mathrm{DP}}^2$.

**Assumption E.2.** A sufficiently large clipping $c$ can be selected such that $\|x_{k,i}\|_2 \leq c$ with high probability.

Note that Assumption E.1 primarily serves to make results cleaner and easily interpretable (see Appendix E.5 for extensions). Assumption E.2 is mild since Gaussians have strong tail decay.

## E.4  Omitted Details and Proofs

**Lemma 11 of [68].**  Our analysis (particularly Lemma 6.2) makes use of the following lemma to determine the posterior of an unknown parameter given several independent Gaussian observations.

**Lemma E.3** (Lemma 11 of [68]). *Let $\theta \in \mathbb{R}$ be a constant (non-informative prior). Let $\{\phi_k \triangleq \theta + z_k\}_{k \in [K]}$ with $z_k \sim \mathcal{N}(0, \sigma_k^2)$ be $m$ independent noisy observations of $\theta$ with variances $\{\sigma_k^2\}_{k \in [K]}$. Then, conditioned on $\{\phi_k\}_{k \in [K]}$, with $\sigma_\theta^2 \triangleq (\sum_{k \in [K]} 1/\sigma_k^2)^{-1}$, we have*

$$\theta = \sigma_\theta^2 \sum_{k \in [K]} \frac{\phi_k}{\sigma_k^2} + z, \text{ where } z \sim \mathcal{N}(0, \sigma_\theta^2). \tag{19}$$

This lemma allows us to express the local true centers $w_k$ conditioned on the local empirical estimates $\hat{w}_k$, the external empirical estimates $\hat{w}_{\backslash k}$, and the local datasets $\{X_k\}_{k \in [K]}$.

Below we present omitted proofs for the main results presented in Section §6. We restate the lemmas and theorems for convenience.

**Lemma 6.1**. The minimizer of the silo-specific objective $h_k(w)$ (see Eq. (3), Eq. (18)) is

$$\hat{w}_k(\lambda) = \alpha \cdot \hat{w}_k + (1 - \alpha) \cdot \hat{w}_{\backslash k}, \quad \text{where} \quad \alpha \triangleq \frac{K + \lambda}{(1 + \lambda)K} \in (1/K, 1]. \tag{20}$$

*Proof of Lemma 6.1.* First note that $\bar{w}$ is independent of $\lambda$. Since we assume $F_k$ is convex, $h_k$ is also convex, and the proof follows from taking the derivative of $h_k$ and setting it to 0:

$$\frac{\partial h_k(w)}{\partial w} = \frac{1}{n} \sum_{i=1}^{n} (w - x_{k,i}) + \lambda(w - \bar{w}) = (w - \hat{w}_k) + \lambda(w - \bar{w}), \tag{21}$$

$$\hat{w}_k(\lambda) = \frac{1}{1 + \lambda} \hat{w}_k + \frac{\lambda}{1 + \lambda} \bar{w} = \frac{K + \lambda}{(1 + \lambda)K} \cdot \hat{w}_k + \frac{\lambda(K - 1)}{(1 + \lambda)K} \hat{w}_{\backslash k}. \tag{22}$$

Note here that in the non-private case, the local estimator is given by the empirical average: $\hat{w}_k = \frac{1}{n} \sum_{i=1}^{n} x_{k,i}$. In the private case, under Assumption E.2, the local estimator is $\hat{w}_k = \frac{1}{n} \left( \xi_k + \sum_{i=1}^{n} x_{k,i} \right)$ where $\xi_k \sim \mathcal{N}(0, \sigma_{\mathrm{DP}}^2)$ is the one-shot privacy noise, and we can similarly arrive at the same $\hat{w}_k(\lambda)$ expression. $\square$

**Lemma 6.2**. Let $\sigma_{\mathrm{loc}}^2 \triangleq \sigma^2/n + \sigma_{\mathrm{DP}}^2/n^2$ denote the local variance on each silo as a result of data sampling and DP noise. Then, given $\hat{w}_k$ and $\hat{w}_{\backslash k}$, $w_k$ can be expressed as

$$w_k = \mu_k + \zeta_k \tag{23}$$

$$\text{where} \quad \zeta_k \sim \mathcal{N}(0, \sigma_w^2), \tag{24}$$

$$\text{with} \quad \sigma_w^2 \triangleq \left( \frac{1}{\sigma_{\mathrm{loc}}^2} + \frac{K - 1}{K\tau^2 + \sigma_{\mathrm{loc}}^2} \right)^{-1} \tag{25}$$

$$\text{and} \quad \mu_k \triangleq \sigma_w^2 \left( \frac{1}{\sigma_{\mathrm{loc}}^2} \cdot \hat{w}_k + \frac{K - 1}{K\tau^2 + \sigma_{\mathrm{loc}}^2} \cdot \hat{w}_{\backslash k} \right). \tag{26}$$

*Proof of Lemma 6.2.* When given $\theta$, we can express $w_k = \theta + z_k$ where $z_k \sim \mathcal{N}(0, \tau^2)$, and by symmetry when given $w_k$,

$$\theta = w_k + z_k \quad \text{where} \quad z_k \sim \mathcal{N}(0, \tau^2). \tag{27}$$

Note also that when given the local dataset $X_k$, $\hat{w}_k$ is a noisy observation of $w_k$ with Gaussian noise from data sampling and added privacy:

$$\hat{w}_k = w_k + \hat{z}_k \quad \text{where} \quad \hat{z}_k \sim \mathcal{N}(0, \sigma_{\mathrm{loc}}^2). \tag{28}$$

Moreover, when given the local datasets $\{X_j\}_{j \in [K], j \neq k}$, $\hat{w}_{\backslash k}$ can be viewed as a noisy observation of $\theta$ with Gaussian noise from the silo heterogeneity and the empirical mean of the local estimators:

$$\hat{w}_{\backslash k} = \theta + \hat{z}_{\backslash k} \quad \text{where} \quad \hat{z}_{\backslash k} \sim \mathcal{N}\left( 0, \frac{\tau^2 + \sigma_{\mathrm{loc}}^2}{K - 1} \right). \tag{29}$$

Combining (29) with (27) we have, when given $w_k$ and the local datasets $\{X_j\}_{j \in [K], j \neq k}$,

$$\hat{w}_{\backslash k} = w_k + z_{\backslash k} \quad \text{where} \quad z_{\backslash k} \sim \mathcal{N}\left( 0, \tau^2 + \frac{\tau^2 + \sigma_{\mathrm{loc}}^2}{K - 1} \right) \equiv \mathcal{N}\left( 0, \frac{K\tau^2 + \sigma_{\mathrm{loc}}^2}{K - 1} \right). \tag{30}$$

Invoking Lemma E.3 on Eq. (28) and Eq. (30) we have the desired $\mu_k$ and $\sigma_w^2$.

A key observation for this derivation is that the local datasets $\{X_k\}_{k \in [K]}$ form the Markov blankets of $\hat{w}_k$ and $\hat{w}_{\backslash k}$ (as they are sampled *after* $w_k$ are sampled, and the estimators $\hat{w}_k$ and $\hat{w}_{\backslash k}$ are computed based on the datasets). Thus, given $\{X_k\}_{k \in [K]}$, $\hat{w}_k$ and $\hat{w}_{\backslash k}$ can be viewed as independent observations of $w_k$ and Lemma E.3 applies. $\square$

**Theorem 6.3** (Optimal MR-MTL estimate). Let $\lambda^*$ be the optimal $\lambda$ that minimizes the generalization error. Then,

$$\lambda^* = \underset{\lambda}{\arg\min}\, \mathbb{E}\left[(w_k - \hat{w}_k(\lambda))^2 \mid \hat{w}_k, \hat{w}_{\backslash k}, \{X_k\}_{k \in [K]}\right] = \frac{\sigma_{\mathrm{loc}}^2}{\tau^2} = \frac{1}{n\tau^2}\left(\sigma^2 + \frac{\sigma_{\mathrm{DP}}^2}{n}\right). \quad (31)$$

*Proof of Theorem 6.3.* The objective of finding $\lambda^*$ equates to finding an expression for $\lambda$ such that when $w_k$ is viewed as a noisy observation of $\hat{w}_k(\lambda)$ (which is an interpolation of $\hat{w}_k$ and $\hat{w}_{\backslash k}$ from Lemma 6.1), the coefficients for $\hat{w}_k$ and $\hat{w}_{\backslash k}$ in $\hat{w}_k(\lambda)$ matches those of $w_k$ (Lemma 6.2). That is, we want $\lambda^*$ such that

$$\frac{K + \lambda^*}{(1 + \lambda^*)K} = \frac{\sigma_w^2}{\sigma_{\mathrm{loc}}^2} = \frac{1}{\sigma_{\mathrm{loc}}^2} \cdot \left(\frac{1}{\sigma_{\mathrm{loc}}^2} + \frac{K-1}{K\tau^2 + \sigma_{\mathrm{loc}}^2}\right)^{-1}. \quad (32)$$

Simpifying gives $\lambda^* = \sigma_{\mathrm{loc}}^2/\tau^2$. Note that $\hat{w}_k(\lambda^*)$ is the (conditional) MMSE estimator of $w_k$. $\square$

Note that Eq. (31) measures the *generalization* error because $w_k$ is the (unobserved) true center of the local data distribution around which testing data points would be drawn.

**Proposition 6.5** (Optimal error gap to local training). Denote the error of the local estimate as

$$\mathcal{E}_{\mathrm{loc}} \triangleq \mathbb{E}\left[(w_k - \hat{w}_k)^2 \mid X_k\right], \quad (33)$$

and let $\Delta_{\mathrm{loc}} \triangleq \mathcal{E}_{\mathrm{loc}} - \mathcal{E}^*$ be its error gap to the optimal estimate. Then,

$$\Delta_{\mathrm{loc}} = \left(1 - \frac{1}{K}\right) \cdot \frac{\sigma_{\mathrm{loc}}^4}{\sigma_{\mathrm{loc}}^2 + \tau^2}. \quad (34)$$

**Proposition 6.6** (Optimal error gap to FedAvg). Denote the error of the global (FedAvg) estimate as

$$\mathcal{E}_{\mathrm{fed}} \triangleq \mathbb{E}\left[(w_k - \bar{w})^2 \mid \{X_k\}_{k \in [K]}\right], \quad (35)$$

and let $\Delta_{\mathrm{fed}} \triangleq \mathcal{E}_{\mathrm{fed}} - \mathcal{E}^*$ be its error gap to the optimal estimate. Then,

$$\Delta_{\mathrm{fed}} = \left(1 - \frac{1}{K}\right) \cdot \frac{\tau^4}{\sigma_{\mathrm{loc}}^2 + \tau^2}. \quad (36)$$

*Proofs of Propositions 6.5 and 6.6.* From Corollary 6.4 we know that the error of the optimal estimator $\hat{w}_k(\lambda^*)$ of $w_k$ is $\mathcal{E}^* = \sigma_w^2 = \frac{\sigma_{\mathrm{loc}}^2(\sigma_{\mathrm{loc}}^2 + K\tau^2)}{K(\sigma_{\mathrm{loc}}^2 + \tau^2)}$. For $\mathcal{E}_{\mathrm{loc}}$, we know from earlier that the local estimator $\hat{w}_k = \hat{w}_k(0)$ of $w_k$ has an MSE/variance of $\mathcal{E}_{\mathrm{loc}} = \sigma_{\mathrm{loc}}^2$. For $\mathcal{E}_{\mathrm{fed}}$, we can view $\bar{w}$ as a noisy observation of $w_k$ with MSE/variance $\mathcal{E}_{\mathrm{fed}} = \frac{\sigma_{\mathrm{loc}}^2 + (K-1)\tau^2}{K}$. The results of Propositions 6.5 and 6.6 thus follow from re-arranging terms of $\mathcal{E}_{\mathrm{loc}} - \mathcal{E}^*$ and $\mathcal{E}_{\mathrm{fed}} - \mathcal{E}^*$ respectively. $\square$

**Lemma 6.7** (Error of $\hat{w}_k(\lambda)$). Let $\mathcal{E}_\lambda \triangleq \mathbb{E}\left[(w_k - \hat{w}_k(\lambda))^2 \mid \hat{w}_k, \hat{w}_{\backslash k}, \{X_k\}_{k \in [K]}\right]$ be the error of MR-MTL as a function of $\lambda$. Then,

$$\mathcal{E}_\lambda = \left(1 - \frac{1}{K}\right) \frac{\sigma_{\mathrm{loc}}^2 + \lambda^2\tau^2}{(\lambda + 1)^2} + \frac{\sigma_{\mathrm{loc}}^2}{K}. \quad (37)$$

*Proof of Lemma 6.7.* The result follows from the variance of $\hat{w}_k(\lambda)$ when viewed as a noisy observation of $w_k$. Specifically, from Lemma 6.1 we know that $\hat{w}_k(\lambda)$ is an interpolation (parameterized by $\alpha \triangleq \frac{K+\lambda}{(1+\lambda)K}$) between $\hat{w}_k$ and $\hat{w}_{\backslash k}$. We thus have (with some rearrangement),

$$\mathcal{E}_\lambda = \alpha^2 \cdot \sigma_{\mathrm{loc}}^2 + (1-\alpha)^2 \cdot \frac{K\tau^2 + \sigma_{\mathrm{loc}}^2}{K-1} = \left(1 - \frac{1}{K}\right) \frac{\sigma_{\mathrm{loc}}^2 + \lambda^2\tau^2}{(\lambda + 1)^2} + \frac{\sigma_{\mathrm{loc}}^2}{K}. \quad (38)$$

$\square$

The proof of Theorem 6.8 follows directly from Lemma 6.7 by subtracting the common terms between private and non-private MR-MTL estimates.

### E.5 The Case of Heterogeneous Privacy Requirements

In the main analysis, we assumed for simplicity that each silo has the same values of local dataset size $n$, local data sampling variance $\sigma^2$, and DP noise variance $\sigma_{\mathrm{DP}}^2$. In this section, we consider the

case where each silo $k$ has custom values for the above, denoted as $n_k$, $\sigma_k^2$, and $\sigma_{k,\mathrm{DP}}^2$, to arrive at a silo-specific "local variance" $\tilde{\sigma}_k^2$:

$$\tilde{\sigma}_k^2 \triangleq \frac{\sigma_k^2}{n} + \frac{\sigma_{k,\mathrm{DP}}^2}{n^2}. \tag{39}$$

With a slight abuse of notation, let us use $C + \mathcal{N}(\mu, \sigma^2)$ to denote drawing an iid sample of Gaussian noise with mean $\mu$ and variance $\sigma^2$ and add it to some value $C$. Then, following Lemma 6.2 and the same set of conditions, we can express the local model $\hat{w}_k$ and the non-local model $\hat{w}_{\backslash k}$ as:

$$\hat{w}_k = w_k + \mathcal{N}(0, \tilde{\sigma}_k^2), \tag{40}$$

$$\hat{w}_{\backslash k} = \theta + \mathcal{N}\left(0, \frac{\tau^2}{K-1} + \frac{\sum_{j \neq k} \tilde{\sigma}_j^2}{(K-1)^2}\right) \tag{41}$$

$$= w_k + \mathcal{N}\left(0, \tau^2 + \frac{\tau^2}{K-1} + \frac{\sum_{j \neq k} \tilde{\sigma}_j^2}{(K-1)^2}\right) \tag{42}$$

$$= w_k + \mathcal{N}\left(0, \frac{K\tau^2}{K-1} + \frac{\sum_{j \neq k} \tilde{\sigma}_j^2}{(K-1)^2}\right), \tag{43}$$

and the true local center $w_k$ can be expressed in terms of $\hat{w}_k$ and $\hat{w}_{\backslash k}$ with silo-specific local variances:

**Lemma E.4.** *Given $\hat{w}_k$ and $\hat{w}_{\backslash k}$, $w_k$ can be expressed as*

$$w_k = \bar{\mu}_k + \mathcal{N}(0, \bar{\sigma}_k^2) \tag{44}$$

*with*

$$\bar{\sigma}_k^2 \triangleq \left(\frac{1}{\tilde{\sigma}_k^2} + \frac{(K-1)^2}{(K-1)K\tau^2 + \sum_{j \neq k} \tilde{\sigma}_j^2}\right)^{-1}, \tag{45}$$

$$\bar{\mu}_k \triangleq \frac{\bar{\sigma}_k^2}{\tilde{\sigma}_k^2} \cdot \hat{w}_k + \frac{(K-1)^2 \bar{\sigma}_k^2}{(K-1)K\tau^2 + \sum_{j \neq k} \tilde{\sigma}_j^2} \cdot \hat{w}_{\backslash k}. \tag{46}$$

*Proof.* The proof follows similarly from Lemma 6.2: we apply Lemma E.3 to the expressions of $\hat{w}_k$ and $\hat{w}_{\backslash k}$ as noisy observations of $w_k$. $\qquad\square$

From here we can derive the best $\lambda_k^*$ specific to each silo.

**Theorem E.5** (Optimal $\lambda$ for silo $k$). *The optimal choice of the MR-MTL regularization strength for silo $k$ is given by*

$$\lambda_k^* = \frac{\tilde{\sigma}_k^2}{\tau^2 + \frac{1}{K}\left(\frac{\sum_{j \neq k} \tilde{\sigma}_j^2}{K-1} - \tilde{\sigma}_k^2\right)} \tag{47}$$

*Proof.* The result follows from re-arranging and solving for $\lambda_k^*$ in the following equation, as in Theorem 6.3:

$$\frac{K + \lambda_k^*}{(1 + \lambda_k^*)K} = \frac{\bar{\sigma}_k^2}{\tilde{\sigma}_k^2} \tag{48}$$

where $\mu_k$ is defined in Lemma E.4. $\qquad\square$

Theorem E.5 suggests that if silos have varying local variance (as a result of varying local dataset sizes, inherent data variances, and silo-specific sample-level DP requirements), then it is optimal to have each silo choose its own $\lambda_k^*$. In particular, the term $\frac{\sum_{j \neq k} \tilde{\sigma}_j^2}{K-1} - \tilde{\sigma}_k^2$ suggests that the optimal $\lambda_k^*$ depends on how much of an "outlier" is silo $k$ compared to the rest of silos: if it has a smaller local variance, then it benefits from a smaller $\lambda_k^*$ since other silos are noisy; if it has a larger local variance, then $\lambda_k^*$ would be larger to encourage silo $k$ to "conform". Note that under this simple analysis, one could have $\lambda_k^* < 0$ (when $\tilde{\sigma}_k^2 > K\tau^2 + \frac{1}{K-1}\sum_{j \neq k} \tilde{\sigma}_j^2$); this case stems from having a gradually larger $\tilde{\sigma}_k^2$ and it suffices to choose $\lambda^* \to \infty$ (i.e. fall back to FedAvg training).

# F   Additional Details for Private Hyperparameter Tuning (§7)

In Section §7 we took an alternative view on the benefits of personalization by examining the *privacy cost* of tuning the hyperparameter responsible for navigating the tradeoff between costs from heterogeneity and privacy. In this section, we expand on the omitted details and provide more discussions.

## F.1   Additional Background on Privacy Costs of Hyperparameter Tuning

We consider the typical hyperparameter tuning procedure (denoted as HPO) where a base algorithm $M$, such as one run of a machine learning algorithm to produce a model, is executed $h$ times with different hyperparameters and the best result is recorded. The hyperparameters may be high-dimensional; e.g. vector of learning rate and batch size. Typically, $h$ is a constant (e.g. we sweep a fixed grid of learning rates and batch sizes).

On a high level, the works of Liu and Talwar [62] and Papernot and Steinke [79] show that, with a constant $h$, one can construct $M$ where the privacy guarantee of the HPO output is (roughly) a factor of $h$ weaker than the original privacy guarantee. Specifically, [79] gives the following proposition that applies to both pure DP and Rényi DP (and thus related to approximate DP).

**Proposition F.1** (Proposition 17 of [79])**.** *For any $\varepsilon > 0$ and any $\alpha > 1$, there exist some algorithm $M$ that satisfy $(\varepsilon, 0)$-DP and the corresponding HPO algorithm $A$ that runs $M$ for $h$ times and returns only the best result, such that*

1. *$A$ is not $(\tilde{\varepsilon}, 0)$-DP for any $\tilde{\varepsilon} < h\varepsilon$, and*

2. *$A$ is not $(\alpha, \tilde{\varepsilon}(\alpha))$-Rényi DP for any $\tilde{\varepsilon}(\alpha) < \hat{\varepsilon}(\alpha)$ where*

$$\hat{\varepsilon}(\alpha) = h\varepsilon - \frac{h \log(1 + \exp(-\varepsilon))}{\alpha - 1}. \tag{49}$$

Both the pure DP and RDP results suggest that running the typical HPO procedure with a fixed $h$ can be as bad as running $M$ and $h$ times and releasing *all* results.

The authors of [62] and [79] provided a mitigating strategy where one simply makes $h$ (the number of tuning runs) a random variable. The authors of [79] provide an improved analysis and considered two distributions for $h$ in particular: the truncated negative binomial (TNB) distribution and the Poisson distribution.

Here, we focus on TNB as it provides a spectrum of results for privacy-utility tradeoff, allowing us to choose several points on this spectrum that offer small (but realistic) privacy overhead. We combine it our analysis in Section §6 to motivate our discussions on the practical complications of deploying personalization methods.

Specifically, the probability mass function of the TNB distribution is given by

$$f(h) = \begin{cases} \frac{(1-\gamma)^h}{\gamma^{-\eta}-1} \prod_{\ell=0}^{h-1} \left( \frac{\ell+\eta}{\ell+1} \right) & \text{for } \eta \neq 0, \\ \frac{(1-\gamma)^h}{h \log(1/\gamma)} & \text{for } \eta = 0, \end{cases} \quad \text{with expected value } \mathbf{E}[h] = \begin{cases} \frac{\eta(1-\gamma)}{\gamma(1-\gamma^\eta)} & \text{for } \eta \neq 0, \\ \frac{1/\gamma-1}{\log(1/\gamma)} & \text{for } \eta = 0, \end{cases}$$

where $\eta$ and $\gamma$ are parameters of the TNB distribution. The privacy of the HPO output using $h$ sampled from the TNB distribution is given by the following theorem from [79].

**Theorem F.2** (Theorem 2 of [79])**.** *For any $\eta > 1$, any $\gamma \in (0,1)$, and any algorithm $M$ that satisfy both $(\alpha_1, \varepsilon(\alpha_1))$-RDP and $(\alpha_2, \varepsilon(\alpha_2))$-RDP with $\alpha_1 > 1$ and $\alpha_2 \geq 1$, the HPO algorithm $A$ that runs $M$ for $h$ times and returns only the best result with $h \sim \text{TNB}(\eta, \gamma)$ satisfies $(\alpha_1, \tilde{\varepsilon})$-RDP with*

$$\tilde{\varepsilon} = \varepsilon(\alpha_1) + (1+\eta)\left(1 - \frac{1}{\alpha_2}\right)\varepsilon(\alpha_2) + \frac{(1+\eta)\log(1/\gamma)}{\alpha_2} + \frac{\log \mathbf{E}[h]}{\alpha_1 - 1}. \tag{50}$$

Theorem F.2 tells us that by drawing $h$ from the TNB distribution, the privacy cost of the HPO procedure is a *constant* factor of the original cost of $M$ for pure DP (when $\alpha_1, \alpha_2 \to \infty$), or up to a factor of $\log \mathbf{E}[h]$ for approximate DP. This substantially improves the default HPO procedure.

## F.2   Interpretation of Fig. 7

In Fig. 7, we considered 6 pairs of $(\eta, \gamma)$ to arrive at the same $\mathbf{E}[h] = 10$ (i.e. on average we want to try 10 values of $\lambda$ when tuning MR-MTL). Intuitively, for a fixed $\mathbf{E}[h]$, we can expect a smaller $\eta$ (and consequently a smaller $\gamma$) to give a tighter privacy guarantee from Theorem F.2 (i.e. the privacy cost of hyperparameter tuning is smaller). At the same time, however, the TNB distribution would be more concentrated around $h = 1$ with a smaller $\eta$, which means we may end up trying only 1 hyperparameter even when $\mathbf{E}[h]$ is large. $(\eta, \gamma)$ thus provides an implicit privacy-utility tradeoff.

In Fig. 7 we chose a list of $\eta$ values that favors strong privacy while being realistic, since picking $\eta \to -1$ (or just $\eta < -0.5$) would give a even smaller privacy overhead, but it will perform very poorly utility-wise and was not considered by [79]. We then applied Theorem F.2 with these $(\eta, \gamma)$ values to the the expected error of MR-MTL as a function of $\lambda$ under mean estimation, by computing and applying the extra noise needed to account for the privacy cost of tuning. Here, the closed form of the expected error of MR-MTL is derived in Lemma 6.7, but in Fig. 7 we ran numerical simulations with 500 repetitions.

Fig. 7 (a) and (c) suggests that there are settings in which, after applying Theorem F.2, the best $\lambda^*$ one could find would already give an error that is larger than simply running FedAvg and local training *without any tuning*, respectively (the **green** curves). Fig. 7 (b) suggests that in less extreme cases, the private hyperparameter tuning cost would significantly diminish the utility improvement of MR-MTL.

## F.3   Implementing Hyperparameter Tuning under Silo-Specific Sample-Level DP

Under silo-specific sample-level privacy, it can be intricate to implement the improved, randomized hyperparameter tuning protocol described above in Appendix F.1. For silo $k$ to satisfy its own $(\varepsilon_k, \delta_k)$ sample-level DP target, it must draw $h$ *independently* to all other silos, and it must only return the best result of hyperparameter tuning at the *end of training* (instead of returning the best model iterate at every round).

Using MR-MTL as an example, the above has several implications:

1. Each silo should now maintain a list of personalized models, each corresponding to a choice of $\lambda$.

2. Depending on the specific distribution of $h$, silos may or may not end up trying the same set of $\lambda$ values.

3. During training rounds, silos must return model updates for – and consequently regularize their personalized models towards – a "pivot" model using some public $\bar{\lambda}$.

The analysis in Appendix E.5 suggests that having silo-specific choices of $\lambda$ (#2 above) should not have adverse effects, but the above adaptations in general may influence final utility or convergence properties in iterative learning settings.

Other more sophisticated implementation (potentially tailored to the specific personalization method) may also be possible. It would be an interesting future direction to study the best approaches for implementing private hyperparameter tuning under silo-specific sample-level privacy.

# G  Future Directions for Auto-tuning / Online Estimation of $\lambda$

Apart from tuning $\lambda$ for MR-MTL (or other hyperparameters for other personalization methods) non-adaptively with grid search or random search, another approach is to leverage some form of online estimation as in [97, 8, 80] (mentioned in Section §7). That is, $\lambda$ is adjusted at every training round or iteration such that it gradually converges to a good $\lambda^*$ (in expectation).

This line of approaches would be promising if their privacy overhead (if any) can be contained within a small factor of the original privacy guarantee—smaller than that offered by the randomized tuning procedure discussed above in Appendix F.1. However, since the hyperparameters of interest are those that navigate the tradeoff between costs from heterogeneity and privacy (such as $\lambda$ for MR-MTL), there are several factors that make auto-tuning particularly challenging in our setting:

- **A universal and practical measure of heterogeneity is generally unavailable**. For MR-MTL, our analysis in Section §6 suggests that the optimal $\lambda^*$ depends on a measure of data heterogeneity across silos ($\tau^2$). However, in practice, the level of heterogeneity is abstract and hard to quantify even with non-private oracle access to the silo datasets. Past work has devised custom measures of heterogeneity such as the variance of client (silo) model or gradient updates [52, 59], or the difference between the true global loss and the aggregate of the true local losses, but these measures may not be consistent throughout training [52, 59] such that they cannot provide a useful estimate of the heterogeneity which should be invariant during training. These metrics may also simply not be obtainable in practice [60]. The work of [91] considers the use of a client-relationship matrix, but such matrix may be too large ($K^2$ elements) and too noisy under privacy as we may need to directly or implicitly noise every coordinate of the matrix. Directly estimating the dataset statistics may also be challenging, since silos may apply custom dataset transformations (e.g. standardization or augmentation) that can drastically alter such statistics without affecting the learning dynamics.

- **The estimated heterogeneity may have high variance.** In addition to the above, heterogeneity would generally be measured on the client distribution (as some form of "variance" across clients, as in [52]), but there is generally a limited number of clients in cross-silo settings, meaning that this estimated "variance" as a measure of heterogeneity may itself have high variance. Moreover, such measures must also be estimated privately, which means that it would have even higher variance from the additional noise. This could be problematic since the best hyperparameter for personalization may depend on such heterogeneity measure (as is the case for MR-MTL in simplified settings), and the precision of a potential auto-tuning precedure may suffer as a result.

- **Auto-tuning procedures may need to be developed specifically for each method.** For example, local finetuning may use the number of federated training rounds as the hyperparameter for determining "how much" to personalize, but such hyperparameter would likely be auto-tuned in a drastically different way compared to auto-tuning for $\lambda$ for MR-MTL. It is in general an open question as to whether method-specific auto-tuning procedures will always outperform the simple randomized procedure described in Appendix F.1, in terms of both precision and privacy overhead.

## H    Additional Experimental Results

### H.1    Local Finetuning (Extension of Fig. 2)

In Fig. 2, we made the observation that local finetuning [99, 104, 19] as a personalization strategy (warm-starting from a global model and continue training using local data) may not always improve utility as expected. Extending on Fig. 2, we examine the behavior of local finetuning starting at different stages of training (under a fixed total number of rounds) as well as under varying privacy budgets in Fig. A2. We observe that the phenomenon illustrated in Fig. 2 is indeed reflected across different settings: as soon as local finetuning begins, the DP utility gap can quickly widen, and the utility of finetuning under privacy can roughly reduce to the utility of local training.

Note, however, that these observations serve to give insights into an interesting behavior of local finetuning under silo-specific sample-level DP and *do not* preclude the possibility that local finetuning can still outperform both local training and FedAvg. Indeed, local finetuning was observed to outperform local training in, e.g., Fig. 3 (d) and Fig. A8.

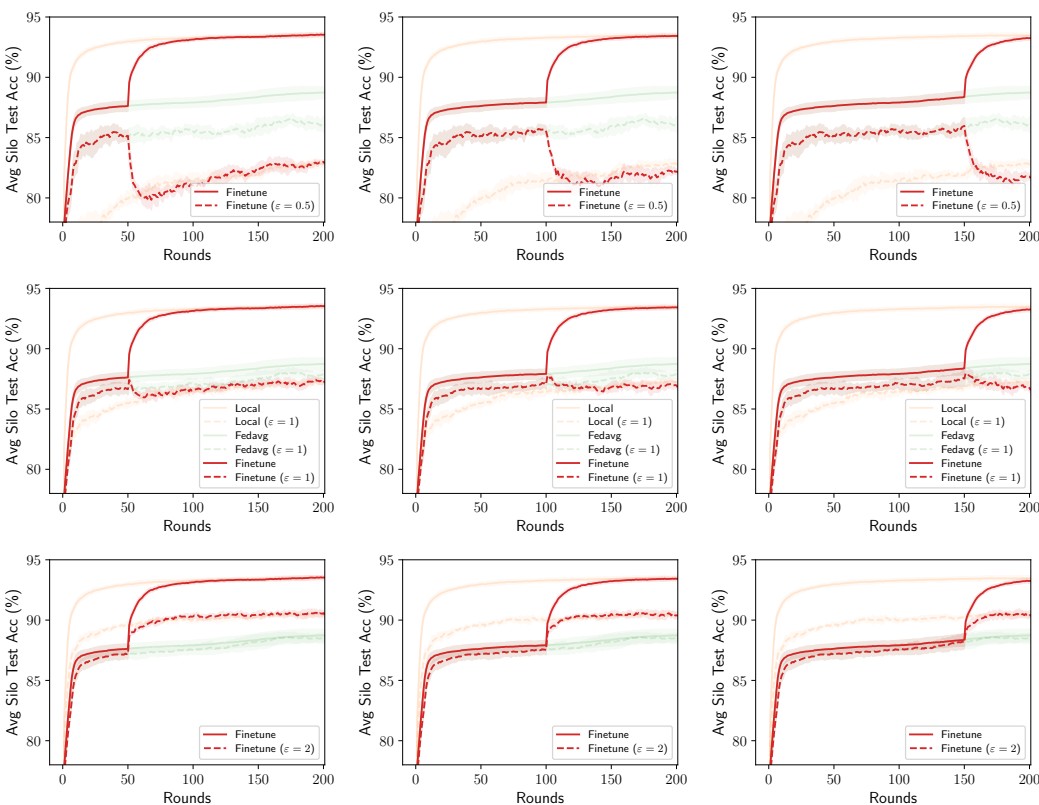

Figure A2: (Extension of Fig. 2) **Behavior of local finetuning** starting at different stages of training (25%, 50%, 75%) and under varying privacy budgets ($\varepsilon \in [0.5, 1, 2], \delta = 10^{-7}$) for $T = 200$ rounds on the **Vehicle** dataset.

### H.2    Utility of MR-MTL as a Function of $\lambda$ (Extension of Fig. 5)

We also extend on Fig. 5 to consider varying privacy budgets, and the results are shown in Fig. A3 (Vehicle), Fig. A4 (School), and Fig. A5 (Heterogeneous CIFAR-10). Note that on School, the test metric is MSE thus lower is better. There are several notable observations:

- $\lambda^*$ **decreases as $\varepsilon$ grows (weaker privacy)**: With weaker privacy (larger $\varepsilon$), we have a smaller optimal $\lambda^*$ (i.e. the location of the utility "bump" under DP gradually shifts to the left). This behavior is characterized by Theorem 6.3, where under stronger privacy, silos benefit from more federation as a means to reduce the effect of privacy noise.

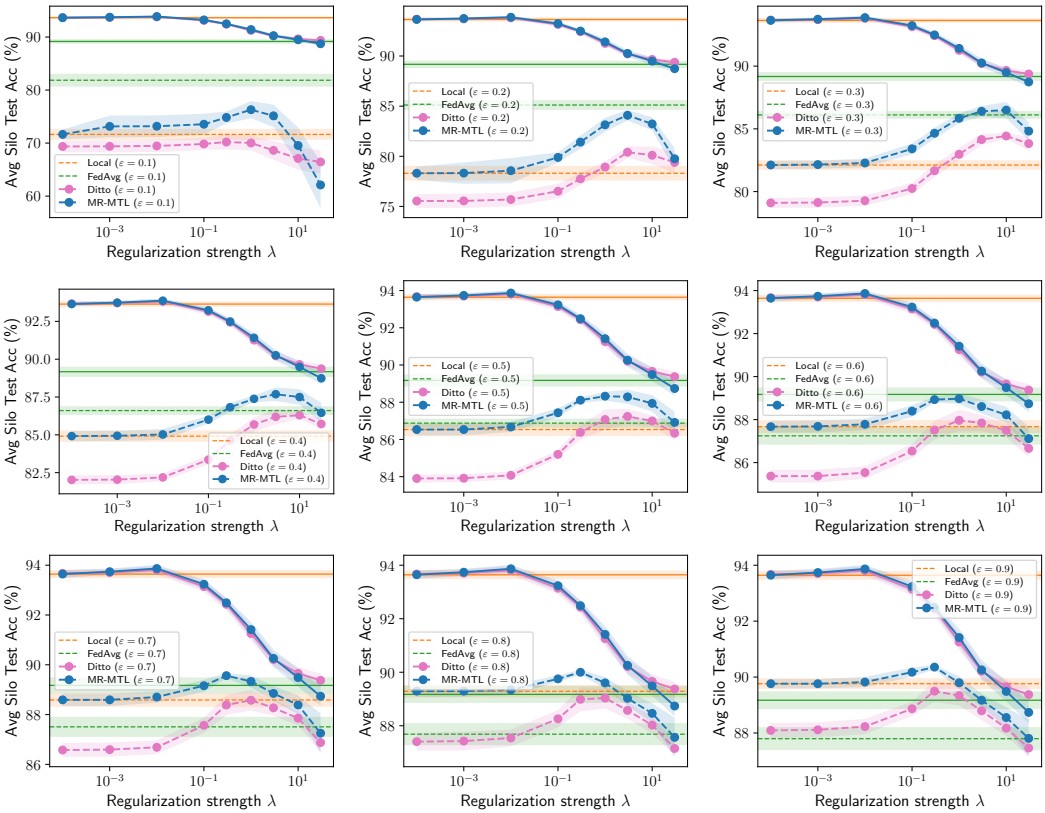

Figure A3: (Extension of Fig. 5) **Behavior of MR-MTL as a function of $\lambda$ across varying privacy budgets** ($\varepsilon \in [0.1, 0.2, ..., 0.9], \delta = 10^{-7}$) for $T = 400$ rounds on the **Vehicle** dataset. Solid lines refer to the non-private runs (same across all plots).

- **MR-MTL does not always outperform FedAvg** (at high privacy regimes): One minor caveat is that Proposition 6.6 says that the utility gap from MR-MTL to FedAvg is always nonnegative under federated mean estimation; however, as discussed in Sections §5 and §6, MR-MTL does not approach FedAvg with larger $\lambda$ in general learning settings since the MR-MTL objective may become too hard to solve via (DP-)SGD. Thus, despite its utility advantage over local training at $\lambda^*$, it may never reach the performance of FedAvg. See, e.g., subplots of $\varepsilon \in [0.1, 0.2]$ in Fig. A3.

- **Utility advantage of MR-MTL ($\lambda^*$) over local/FedAvg changes with $\varepsilon$:** Observe that with larger $\varepsilon$, the utility advantage of MR-MTL ($\lambda^*$) is larger compared to FedAvg and is smaller compared to local training. This behavior is characterized by Propositions 6.5 and 6.6.

Note also that for Heterogeneous CIFAR-10 (Fig. A5), Ditto [57] exhibits a slightly different interpolation behavior compared to MR-MTL (e.g. it has larger optimal $\lambda^*$ under privacy), though at $\lambda^*$ its utility underperforms that of MR-MTL under privacy.

### H.3 Subsampled ADNI Dataset

We further evaluate the suite of personalization methods on the subsampled ADNI dataset, and the results are shown in Fig. A6. There are several notable observations:

1. MR-MTL is essentially recovering local training due to the high degree of heterogeneity present in this dataset, with $\lambda^* \approx 0$ for most $\varepsilon$ values except around $\varepsilon \approx 3.5$ (annotated with arrows in Fig. A6 (a, b)).

2. Unlike results on other datasets, the privacy regime of interest where MR-MTL outperforms both local/FedAvg (around $\varepsilon \approx 4$) is extremely narrow, compared to $\varepsilon \approx 0.5$ for Vehicle (Fig. 3 (a)), $\varepsilon \approx 6$ for School (Fig. 3 (b)), and $\varepsilon \approx 1.5$ for GLEAM (Fig. 3 (c)). The utility advantage of MR-MTL is also insignificant.

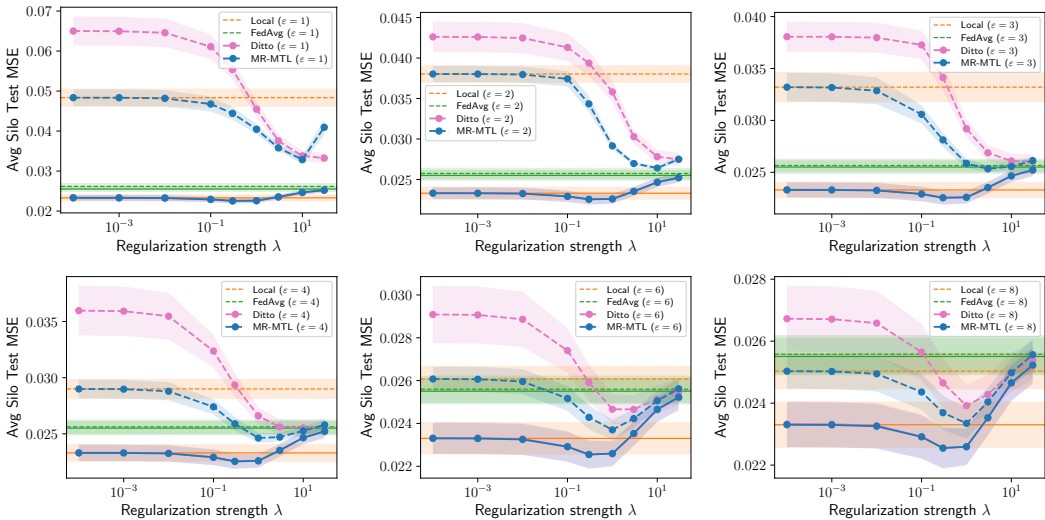

Figure A4: (Extension of Fig. 5) **Behavior of MR-MTL as a function of $\lambda$ across varying privacy budgets** ($\varepsilon \in [1, 2, 3, 4, 6, 8], \delta = 10^{-3}$) for $T = 200$ rounds on the **School** dataset. Cf. Fig. 3 (b) and Fig. A8, where the privacy regime of interest is around $\varepsilon \approx 6$. Solid lines refer to the non-private runs (same across all plots).

3. The underlying heterogeneity structure of the dataset is rather pathological in that it is not ameanable to *both* mean-regularization and clustering. First, observe from Fig. A6 (a) that MR-MTL still opts for a small $\lambda^*$ in high-privacy regimes even when FedAvg performs better; one would expect MR-MTL to opt for a larger $\lambda$ for lower DP noise (at a cost of higher heterogeneity). Second, from Fig. A6 (b, c), we observe that both clustering and cluster-preconditoning did not lead to significant utility improvements, although the latter reduces the degree of heterogeneity and allows MR-MTL to opt for a larger $\lambda^*$ in the privacy regime of interest.

In general, determining how to better model such heterogeneity (especially in high privacy regimes) is an interesting direction of future work.

### H.4 Additional Discussions

**Behavior on larger privacy budgets (Fig. A7).** We extend the results on Vehicle (Fig. 3 (a, b)) by considering larger privacy budgets, and the results are shown in Fig. A7. We note that as privacy requirement loosens, personalized learning (where each silo maintains its own model) tend to perform better than the case where models are shared across clients (FedAvg and IFCA). This is expected as clients in cross-silo datasets tend to have sufficient data for learning a good local model. Moreover, this also means MR-MTL remains competitive as it can use a smaller $\lambda$.

**Effect of dataset subsampling (Fig. A8).** Subsampling local datasets would in principle turn the cross-silo learning setting closer to a cross-device learning setting, in which local training becomes less attractive as due to insufficient local data and may opt for federated training despite data heterogeneity. In Fig. A8, we examine this behavior on the School dataset by using different train/test split ratios of 80%/20%, 50%/50%, and 20%/80%. We observe that with less local training data: (1) FedAvg performs better since silos benefit from federation despite data heterogeneity; (2) the cross-over point between local training and FedAvg shifts to larger $\varepsilon$ (or they may not be a cross-over point); and (3) MR-MTL may no longer provide an optimal point on the personalization spectrum, since small local datasets with silo-specific sample-level DP necessitate a larger $\lambda$ (cf. Theorem 6.3) but MR-MTL may not recover FedAvg under (DP-)SGD. In these cases, client-level DP protection (as is commonly used for cross-device FL) may be more appropriate.

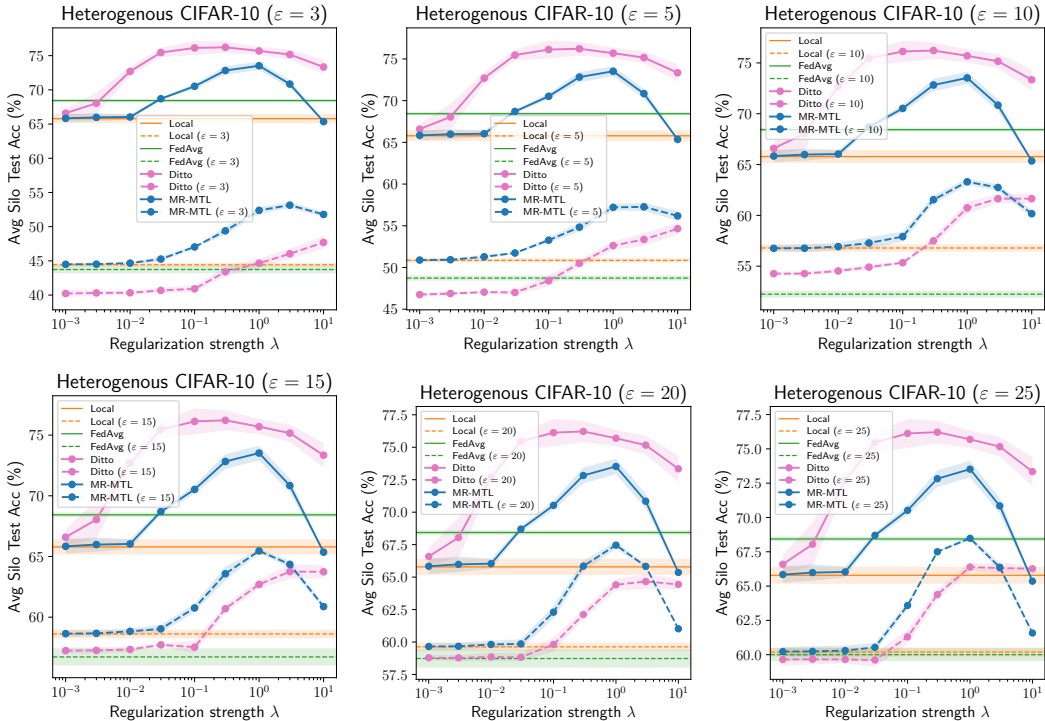

Figure A5: (Extension of Fig. 5) **Behavior of MR-MTL as a function of $\lambda$ across varying privacy budgets** ($\delta = 10^{-4}$) for $T = 200$ rounds on **Heterogeneous CIFAR-10**. Solid lines refer to non-private runs (same across all plots) and dashed/dotted lines refer to private runs (with silo-specific sample-level DP).

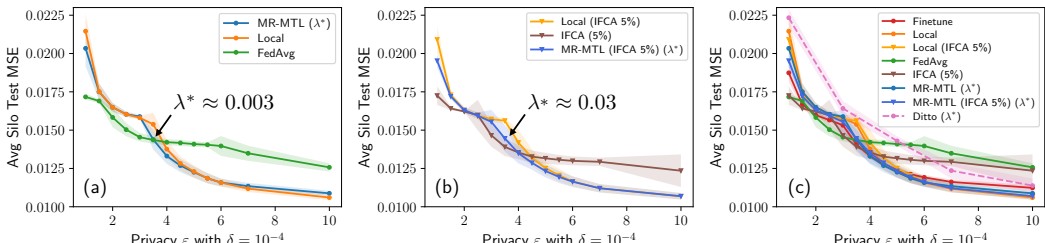

Figure A6: **Test accuracy** (mean $\pm$ std with 3 seeds) vs **privacy budgets** $\varepsilon$ for various personalization methods on the **subsampled ADNI** dataset with $T = 500$ rounds. The heterogeneity present in this dataset is unique in that it is not ameanable to both mean-regularization (a) and clustering (b, c), though clustering can mitigate the level of heterogeneity by allowing a larger $\lambda^*$ for MR-MTL. In such cases of high heterogeneity, local performance tends to be superior for most privacy budgets, and we find that MR-MTL recovers this behavior. Determining how to better model such heterogeneity (especially in high privacy regimes) is an interesting direction of future work.

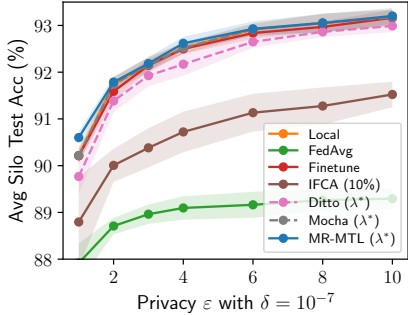

Figure A7: (Extension of Fig. 3 (b)) Results on the **Vehicle** dataset for low privacy regimes ($1 \le \varepsilon \le 10$). As privacy becomes weaker, the need for federation diminishes and personalization methods (including local training) perform better. MR-MTL can recover local training by using a small $\lambda$.

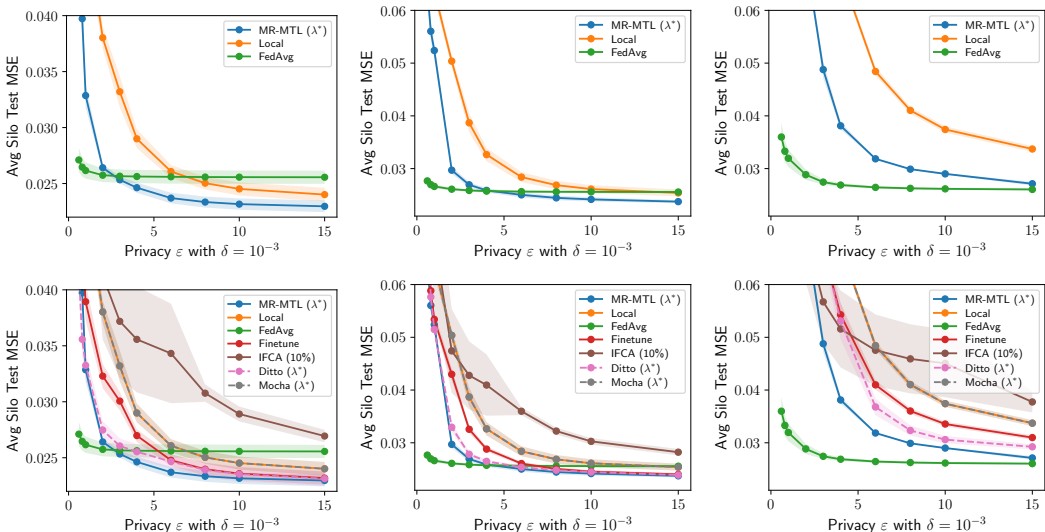

Figure A8: **Test MSE vs total privacy budgets on the School dataset** (80%/20%, 50%/50%, and 20%/80% train/test split of the local dataset by columns from left to right). The top row compares MR-MTL to local training/FedAvg, which form the endpoints of the personalization spectrum with constant privacy costs, and the bottom row compares against other personalization methods. Under data subsampling (as little as 20% training data in the 3rd column), we obtain a setting closer to cross-device FL where FedAvg outperforms personalization since clients benefit from others' training data despite their heterogeneity.