# OpenReview forum: "On Privacy and Personalization in Cross-Silo Federated Learning"
_NeurIPS.cc/2022/Conference — NeurIPS 2022 Accept_

### Official Review · Reviewer_7hbo · 2022-07-06

**Rating:** 6
**Confidence:** 4
**Soundness:** 3 good
**Presentation:** 3 good
**Contribution:** 3 good

**Summary:**

The paper discusses personalization under differential privacy (DP) in federated learning (FL). The key observation in the paper is that with DP, there is a potential trade-off between the amount of federation, that can mitigate DP noise effect, and personalization, which can mitigate problems due to data heterogeneity. The authors then discuss the desiderata for a good personalization approach under DP, and explore the performance of mean-regularized multi-task learning, both theoretically and experimentally.

**Questions:**

1) Reading the current paper, especially lines 24-28, 35-37, 50-51, 109-110, one could be lead to think that this is the first paper to consider sample or item-level DP in cross-silo FL. Please add cites to relevant papers when introducing the problem to remedy this.

2) Related to the previous point, some existing work is presented in a manner which seems a bit misleading, especially w.r.t. the sample-level vs client-level privacy: both Truex et al. 2019 and Heikkilä et al. 2020 define neighbourhood on a single-sample level, not on client-level (see Def.1, and first paragragh in Sec.3, respectively) as claimed. The secure primitives in these papers help to reduce the effect of DP noise, they do not change the basic DP neighbourhood definition used. I think this points to a misunderstanding in the client-level DP definition (see next questions as well).

3) Line 105: "Client-level DP canonically defines a shared privacy guarantee for all participants." Client-level vs sample-level privacy typically relates to the granularity of the DP guarantee, i.e., to the neighbourhood definition used, not having a joint vs individual DP budgets for each client (and incidentally, by tuning the granularity one can then also easily define the individual level privacy, where a single individual may have one or several samples in one silo, that sits between silo-level and sample-level). There is no problem in enforcing client-level DP with varying privacy budgets for different clients in FL (e.g., just use different amounts of noise for different clients with parameter perturbation).

4) Related to the previous comment, on lines 117-120 you claim that client-level DP requires some trust to the central server. This is simply not true, there is no problem in each client enforcing client-level DP individually (although the model utility could be awful), this is local DP (LDP) on the client-level, and by post-processing immunity and parallel composition the global model will have DP guarantees as well, including against the central server.

5) Again a related point, lines 122-123 "local training is 'free' under client-level DP." I would claim that this depends on the DP mechanism, not on the neighbourhood definition: e.g. using DP-SGD and client-level LDP is a counter-example where local training is not free with client-level DP.

6) Lines 33-34: "...participation in FL [is] disclosed publicly, nullifying any client-level DP guarantees." Also lines 121-123. I do not understand why knowledge of participation in FL training would nullify DP guarantees.

7) Lines 166-169: how do you choose the total number of iterations for the fair comparisons?

8) Lines 175-176: if you do not even try to tune the hyperparameters properly for all methods, why should I believe that the results are not simply due to chance?

### Minor comments:

9) Figure 6: due to the scale, it is next to impossible to see from the figure what happens around eps=1-2 that is usually the most important region. Please fix this (change scale, add additional plot maybe to appendix or something else).


10) Lines 86,171: DP-SGD was developed and analysed by Song et al. 2013 and Bassily et al. 2014, while the main contribution of Abadi et al. 2016 was the privacy accounting based on cumulants (moments accountant).

11) DP def. 2.1: the definition of neighbourhood seems a bit strange, since x,x' in mathcal X^n explicitly declares the same dataset size for both, but then you mention addition and removal as well. Which one do you actually use?

References:
Abadi et al 2016: Deep learning with DP.
Bassily et al. 2014: Private empirical risk minimization.
Heikkilä et al. 2020: DP cross-silo FL.
Song et al. 2013: Stochastic gradient descent with DP updates.
Truex et al. 2019: A hybrid approach to privacy-preserving FL.

**Limitations:**

The paper has a good discussion on some important limitations of the proposed method. There is some separate discussion on the potential negative societal impact in the appendix.


**Strengths And Weaknesses:**

### Update after the discussions

I raise my score to recommend acceptance. In case the paper is still not accepted, I would give it a higher score in the future if the following are addressed: 1) make further improvements in the main content (e.g. how to choose lambda under DP, run still more experiments to check how robust the current results are, especially are there cases which go against the current understanding), 2) improve writing clarity (see the discussions, give less weight to sample-level DP discussion and more to the utility trade-off results)

### Strenghts

+ The observation about a possible trade-off between mitigating utility loss due to DP noise by federation and mitigating utility loss due to data heterogeneity by personalization is a nice one and, to my knowledge, a novel perspective on the problem.
+ The theoretical analysis, although under simplifying assumptions, seems to support the argument nicely.
+ The discussion about the price of addional hyperparameters under DP points to an important problem.
+ The paper is mostly clearly written and easy to read.


### Weaknesses

- There seems to be some possible misunderstandings about (client-level) DP in FL, which leads to a number of strange claims in various parts of the paper (see questions for authors for details).

---

> ### Author Response · Authors · 2022-08-02
> **Response to reviewer 7hbo, part 2/2**
>
> (continued from part 1)
>
> Q7
> - For all methods, we first pick a fixed number of FL rounds where **all methods can converge** (e.g. 400 for vehicle).
> - Then, starting from 1 local epoch per round for all methods (which is also a standard choice across many previous works; e.g. [a]), we add extra local epochs/iterations **only when necessary or when there is strong evidence that doing so would improve performance**.
> - This is because privacy cost directly correlates to the overall total number of local gradient steps, adding extra local steps **will directly lead to more DP noise** under the same total privacy budget and can significantly hurt convergence, while the benefit of extra local iterations may be insignificant.
> - Indeed, we tuned the local computation for Mocha and Ditto and observed that extra iterations are often not worth the extra DP noise and should in general be avoided.
>
> Q8
> - We’d like to clarify that hyperparameters are indeed tuned properly for all methods and for different datasets.  For example, on top of important hyperparameters like client LR and $\varepsilon$ values, we also tuned values like $\lambda$ (for Ditto, Mocha, MR-MTL), number of local iterations (Q7 above), number of clusters (for IFCA), percentage of clustering rounds (e.g. 5%, 10%), privacy budget for private cluster selection (e.g. Appendix D.3), percentage of fedavg rounds for finetuning (e.g. Appendix H), etc.
> - The intent of L175-176 was **simply to indicate that we avoid introducing too many variables** by over-tuning less important parameters like batch size, server LR, momentum, server/client optimizers, etc. for which there are reasonable defaults (as done in previous works; e.g. [a, e, g]).
> - All experiments are also repeated over multiple random seeds.
> - We’ll clarify the above in the updated version.
>
> Q9 / Q10
> - Thanks for pointing out these issues! We will fix them accordingly.
>
> Q11
> - Thanks for pointing out this issue! In terms of implementation, we use the “replacement” notion – we provided some discussions in Appendix E.1 under “weighted vs unweighted model updates”. We’ll clarify and ensure consistency in the updated version.
>
> We hope that the above addresses your concerns!
>
> Refs:
> - [a] https://arxiv.org/abs/1710.06963
> - [b] https://arxiv.org/abs/2009.10031
> - [c] https://arxiv.org/pdf/2206.02617.pdf
> - [d] https://proceedings.mlr.press/v162/bietti22a.html
> - [e] https://arxiv.org/abs/2204.13650
> - [f] http://dimacs.rutgers.edu/~graham/pubs/papers/pdp.pdf
> - [g] https://arxiv.org/pdf/2003.00295.pdf
> - [h] https://arxiv.org/abs/1712.07557
> - [i] https://arxiv.org/abs/2102.06387

---

> ### Author Response · Authors · 2022-08-02
> **Response to reviewer 7hbo, part 1/2**
>
> We really appreciate the reviewer for their time, their detailed review, and for recognizing the strengths of this work. We understand that the reviewer is primarily concerned about our claims around client-level DP; **we believe that we are in general agreement with the reviewer, and that most issues stem from the clarity of our writing**. We hope to address your questions in the following.
>
> Q1
> - Thank you for the suggestion! We will add citations / update the main text accordingly.
>
> Q2
> - **Re Heikkilä et al. 2020**: Thank you for raising this issue! We noticed this mistake after the submission and have since corrected it.
> - **Re Truex et al. 2019**: We believe that in Truex et al. 2019, Algorithm 1 and the 2nd last paragraph of section 3.2 suggests that noise is **directly added to the client model update** $Q_s$ (with bounded sensitivity), which would provide a **client-level DP** instead of sample-level DP (please also see our response to Q5 below). At the same time, certain parts of Truex et al. 2019 (e.g. Def. 1, Alg. 4) instead suggest that **sample-level DP** is provided (as noise is now added onto clipped individual gradients).
> - Our goal is to abundantly credit early work that may be related to ours, and in doing so we may have had minor omissions – we will revise accordingly in the updated version. Thanks for pointing these out.
>
>
> Q3
> - We agree with the reviewer that “client-level” and “sample-level” inherently differ in the granularity of protection, rather than that the DP protection is customized for each “record in the dataset” (however this “dataset” may be defined). We believe this to be a minor misunderstanding from our writing clarity.
> - By “canonically … shared privacy guarantee”, we mean that when treating the clients as the “dataset”, a **central client-level $(\epsilon, \delta)$-DP guarantee typically** (e.g. as in [a, b, h, i]) protects all “records” (i.e. clients) equally (since one-shot noise is added on the server), as opposed to more recent notions of “per-instance” accounting (e.g. [c, f]).
> - Indeed, we agree with the reviewer that under **client-level *local*-DP** each client can perturb their updates with varying noise levels and achieve their own DP guarantees.
> - We will improve the writing accordingly.
>
>
> Q4
> - We again agree with the reviewer that client-level local-DP does not require trust; the reviewer might have overlooked L120 where we specifically mentioned “**non-local** DP guarantees”. We will revise and make this clearer.
>
> Q5
> - We have two interpretations for the reviewer’s comment (due to “e.g. using DP-SGD and client-level LDP…”) and address them separately.
> - **[Meaning #1: Using DP-SGD to achieve client-level LDP]**
>     - In essence, client-level local-DP (LDP) is treating each client itself as a single “record” in the “dataset” (which is the set of clients), meaning that under FL, clients can apply local perturbations (clip and noise) to their model update **right before** sending it to the server (standard implementation of LDP in FL), and the **local (personalized) model can be kept noise-free**. In other words, by “local training is ‘free’”, we mean that **local training without noise and ever communicating to the server** can be perfectly private under client-level DP; e.g. see also discussions in [d].
>     - Moreover, we argue that **simply running DP-SGD (within each silo where gradients are clipped and noised) does _not_ give client-level LDP in general**, since the trained model (using private gradients from DP-SGD) can still have **unbounded sensitivity**. Moreover, the **interpretation** of the resulting $(\varepsilon, \delta)$ values would also be different – with large local datasets, it is easy to achieve a small $\varepsilon$ under sample-level DP (w/ DP-SGD) using amplification theorems, while under client-level LDP the $\varepsilon$ would not depend on the local dataset size.
> - **[Meaning #2: Using DP-SGD and a separate client-level LDP mechanism *simultaneously*]**
>     - In this case, we definitely agree with the reviewer that local training is not free (due to the DP-SGD steps _before_ applying LDP perturbations and reporting back to the server).
> - Nevertheless, we value the reviewer’s feedback and will try to make our writing clearer.
>
> Q6
> - The reviewer raised a great point. By “nullifying”, we meant in the **semantic** sense that one may know all the entities inside the “dataset” (the set of silos), and thus the “plausible deniability” property provided by DP for any participants may be weakened; we **did not** mean in the **statistical** sense that the output of the mechanism no longer satisfies the DP definition. We agree that this could be misleading and will revise accordingly.
>
> (continued)

---

> > ### Comment · Reviewer_7hbo · 2022-08-09
> > **Thank you for the clarifications**
> >
> > I tend to agree with the authors that the main weakness I mentioned (discussion and claims about DP) is due to writing clarity, and trust that the authors will fix these issues for the next version of the paper. I will therefore raise my score to recommend accepting the paper: I think the main point about the need to balance utility loss from DP noise against the utility loss from data heterogeneity is a solid and novel line of research.
> >
> > Some further minor comments on the rebuttal:
> >
> > > Q2 We believe that in Truex et al. 2019 [use] client-level DP
> >
> > This is correct, as is clear from their Sec 3.2; my earlier claim was due to careless checking of their definitions.
> >
> > > Q5 We have two interpretations for the reviewer’s comment
> >
> > I ment the first one, so a client running DP-SGD on the local dataset and calculating sensitivity using client-level DP (so, e.g., a client uses full local dataset size as batch size, sums the local gradients, clips the sum to enforce bounded norm for the query, and adds noise. A gradient step should now provide client-level DP since changing the client arbitrarily will always have bounded effect due to clipping, although subsampling amplification etc needs more assumptions). But this issue is a minor sideline for the current paper.

---

### Official Review · Reviewer_dzK1 · 2022-07-10

**Rating:** 3
**Confidence:** 4
**Soundness:** 2 fair
**Presentation:** 2 fair
**Contribution:** 2 fair

**Summary:**

This paper proposes a mean-regularized multi-task learning based cross-silo federated learning for privacy and model personalization.
In cross-silo federated learning, each silo has different item-level different privacy budget on its local dataset
The paper empirically shows that local finetuning fails to achieve the high average silo test accuracy (model personalization) with differential privacy noise after FedAvg steps.
To achieve model personalization under data heterogeneity with differnetial privacy, a network is trained with a mean-regularization penalty.
This penalty is controlled by the hyperparameter \lambda, and the authors propose an optimal value of \lambda in \mu-strongly convex scenario with theoretical analysis.


**Questions:**

I wonder that the same regularization effect will hold for CIAFR-10/100.

**Ethics Review Area:**

["I don’t know"]

**Limitations:**

The paper shows an interesting observation in personalized federated learning with guranteeing diffrential privacy. Mean-regularized multi-task learning method has already been used in personalized FL widely, so I think it will be great if the authors prove the lower bounds of convergence in a nonconvex setting.

**Strengths And Weaknesses:**

Strength
* The proposed approach for personalized FL with guaranteeing differential privacy is simple
* The empirical analysis on several simple benchmark datasets is well demonstrated to help readers to understand.

Weakness
* The paper only provides a theoretical analysis for the convex setting, which may not be directly applied to deep learning scenarios. If it is hard to find the optimal \lambda for a nonconvex setting, the authors may need to give empirical evaluation at least for CIFAR-10/100.
* The references [32] in this paper has already analyzed the personalized federated learning with mean-regularization penalty. This paper simply extends this to the case with the concept of differential privacy. For example, the references [26], [32] in this paper proposed personalized FL algorithms with theoretical guarantee in a nonvex setting, and demonstrated the empirical evaluation for harder cases such as CIFAR-10/100. Therefore, the novelty is not enough to satisfy the standard of this venue.

---

> ### Author Response · Authors · 2022-08-02
> **Response to reviewer dzK1, part 1/2**
>
> We thank the reviewer for their time and feedback and for recognizing the strengths of this work. We understand the reviewer is concerned about (1) the lack of non-convex analysis (or alternatively provide more empirical results), (2) that MR-MTL has been analyzed in previous work, and (3) the lack of convergence results. We hope to address these concerns in the following. Please also see [our shared response](https://openreview.net/forum?id=Oq2bdIQQOIZ&noteId=rMtUPaS_fOR) to all reviewers.
>
> ### Weakness 1: analysis of MR-MTL for non-convex settings, or additional results on CIFAR
>
> Per the reviewer’s request, we added results on heterogeneous CIFAR-10 with both privacy-utility tradeoff curves (analogous to Fig. 3) and the $\lambda$ interpolation curves (analogous to Fig. 5) at the top of the Appendix. Please see our shared response for more details.
>
> For general nonconvex problems, we believe that deriving the optimal $\lambda^*$ (extension of Thm 6.3) may entail deriving the closed-form error of a nonconvex optimization problem as a function of $\lambda$ (extension of Lem 6.7) which could be nontrivial, and we acknowledge that this is a limitation of our work (Appendix C.1). We plan to strengthen our evaluation with more empirical results in the updated version.
>
>
> ### Weakness 2: MR-MTL has been proposed and analyzed before / limited novelty
>
> > The references [32] in this paper has already analyzed the personalized federated learning with mean-regularization penalty … For example, the references [26], [32] in this paper proposed personalized FL algorithms with theoretical guarantee in a nonvex setting
>
> We’d like to point out that:
> - While previous work has indeed studied MR-MTL (or the mean-regularized objective) in general, there is in general a lack of work that considers **its intersection with DP**, and most existing analyses focus on specific aspects such as communication (e.g. [32, 33]) or convergence (e.g. [26, 32, 71]) **that are orthogonal to the focus of this work**; for example, the analysis of [26] focuses on convergence and [32] focuses on providing complexity bounds for communication and local oracle calls.
> - Moreover, **constants matter when it comes to deploying differential privacy**. For example, complexity bounds on local computation such as those provided in [32], while valuable for theoretical understanding, would not be as useful for providing practical recommendations under our DP notion; indeed, Fig. 5 illustrates the DP utility cost of a personalization method (Ditto) when there is a **constant factor (2x)** in the number of local iterations compared to MR-MTL.
>
> Overall, we’d like to emphasize that the goal of our work is to holistically study the application of DP in cross-silo FL  and provide useful insights using both empirical and theoretical analyses, and our work is not geared toward proving or improving a specific theoretical guarantee.
>
> > This paper simply extends this to the case with the concept of differential privacy.
>
> We respectfully disagree with the reviewer that extension to DP is trivial.
> - We believe extending existing algorithms to DP, particularly involving **private model personalization**, is an active research area (e.g., see discussions in [a, b] and our response to Weakness #1 from Reviewer JfCt), and in our case, the notion of silo-specific item-level DP is a meaningful relaxation of the commonly studied in client-level DP for private cross-silo FL motivated by real-world applications (see also our response to Limitations from Reviewer JfCt).
> - Moreover, the emerging empirical phenomena under this particular DP notion (e.g. Fig. 2 and Fig. 5) have not been adequately explored in earlier work, and establishing strong baselines requires carefully accounting for the privacy overhead of local computation from multiple aspects (e.g. L189-195, Fig. 4, Appendix D.3 “private cluster selection”, Section 7, Appendix G.3).
> - In line with reviewers TNbE and 7hbo, we argue that picking a suitable privacy notion, making important observations, and providing relevant analyses – even with existing methods – is a valuable contribution to the community.
>
> > Therefore, the novelty is not enough to satisfy the standard of this venue.
>
> While we value the reviewer’s feedback, we respectfully disagree with this conclusion. With our responses above and our shared response addressing limited algorithmic novelty, we hope that the reviewer will reconsider the merits of our work in **comprehensively examining private cross-silo FL** – from considering a more suitable notion of DP (sec. 3) and studying various emerging phenomena, to providing extensive empirical evaluation (sec. 4), analyzing why MR-MTL works as a strong baseline under this particular DP notion (sec. 5, 6), and examining the practical challenges of deploying MR-MTL (and other potentially more sophisticated personalization methods) (sec 7, Appendix G).
>
> (continued)

---

> ### Author Response · Authors · 2022-08-02
> **Response to reviewer dzK1, part 2/2**
>
> (continued from part 1)
>
> ### Limitations / Weakness #3: lack of convergence results
>
> > I think it will be great if the authors prove the lower bounds of convergence in a nonconvex setting.
>
> Thank you for your suggestion! We’d like to emphasize that **convergence properties are not the focus of this work**. Nevertheless, this is a great question and, to our knowledge, lower bounds of convergence for _nonconvex private training_ is a generally open and active area of research. We will explore this in future work.
>
> Refs:
> - [a] https://arxiv.org/pdf/2108.12978.pdf
> - [b] https://proceedings.mlr.press/v162/bietti22a/bietti22a.pdf
> - [26] https://proceedings.neurips.cc/paper/2020/file/24389bfe4fe2eba8bf9aa9203a44cdad-Paper.pdf
> - [32] https://arxiv.org/pdf/2010.02372.pdf
> - [33] https://arxiv.org/pdf/2002.05516.pdf
> - [35] https://arxiv.org/abs/2108.12978
> - [71] https://proceedings.neurips.cc/paper/2020/file/f4f1f13c8289ac1b1ee0ff176b56fc60-Paper.pdf

---

### Official Review · Reviewer_JfCt · 2022-07-10

**Rating:** 4
**Confidence:** 3
**Soundness:** 2 fair
**Presentation:** 2 fair
**Contribution:** 2 fair

**Summary:**

The paper studies differential privacy (DP) in cross-silo (where participants share different attributes of the same data subjects) federated learning (FL). The authors propose a silo-specific item-level privacy (although it is unclear to me what it means and how it is connected to the methods used to achieve this privacy, see Question section) which is more suitable in cross-silo FL mainly because it can allow different silos (organizations/participants) apply their individual privacy policy (i,e. DP budgets). The paper uses FL personalization methods to achieve this privacy, and empirically finds that MR-MTL (encouraging the mean of each local model's weights to be close to the mean of all local models') lead to the best privacy-utility tradeoff. Then the paper justifies this observation with some theoretical analysis on MR-MTL.

**Questions:**

My major concern that needs clarifications is about the definition of proposed silo-specific item-level privacy. First, what is meant by "item"? From definition 2.1, it seems to mean sample. If so, I strongly recommend the authors to call it "sample" instead because the meaning of "item" is quite unclear. Furthermore, it would be helpful if the authors can provide a formal DP definition of the silo-specific item-level privacy like the definition 2.1.

More importantly, if I understand it correctly, the proposed privacy aims to protect data subjects. If that is the goal, shouldn't the privacy guarantee be data-subject-level rather than sample(or so-called item)-level? I find it hard to understand the proposed privacy definition given the motivation of protecting data subjects because what the paper actually protects is samples in each silo rather than the data subjects. Please let me know if I misunderstand anything.

In addition, if the goal is to protect data subjects, I do not understand why letting each silo adds its own noise, instead of protecting the combined datasets (from all silos), can mitigate the privacy risk since each data subject's full information can only be found in the combined datasets. I would imagine if you want to protect data-subject-level privacy, you would need to operate on the combined dataset. Therefore, it seems to me the proposed solution and privacy definition are disconnected from the motivation of "concern in-silo data subjects". Please let me know if I have a high-level misunderstanding.

Minor concerns
- Define "local training"
- Fig.2, the right figure, explain what is meant by the semi-transparent plot line

**Limitations:**

In additions to the limitations shown in Section 7, I would suggest the authors to include a discussion on the proposed privacy definition. What might go wrong if practitioners go with this less stringent definition? What are the trade-offs between this definition and other DP definitions in cross-silo FL? Under which real-world scenarios, practitioners would be most likely to benefit from the proposed definition?

**Strengths And Weaknesses:**

Strength
- Topic and problem are important. Privacy concerns in cross-silo FL are indeed different from cross-device FL. The motivation that different silos might want to enforce different privacy policies/budgets is, in my opinion, reasonable, and it is a practical problem in the real-world cross-silo FL deployments

Weaknesses
- The proposed silo-specific item-level privacy is not clearly defined (see Questions section)
- There is no novelty in the methodology, the paper only uses existing methods
- The experiments only include two datasets, and performance difference is not very significant. It is unclear how solid the conclusion that MR-MTL is the best can be hold in general

---

> ### Author Response · Authors · 2022-08-02
> **Response to reviewer JfCt, part 1/2**
>
> We thank the reviewer for their time and feedback! We’re glad that the reviewer finds our problem of interest important. We hope to address your concerns in detail below.
>
> ### Weakness #1. “The proposed silo-specific item-level privacy is not clearly defined”
>
> > shouldn't the privacy guarantee be data-subject-level rather than sample(or so-called item)-level?
>
> We agree with the reviewer that “data-subject-level DP” would be ideal; however, as we will detail below, this may be challenging to enforce in practice (an active research area) and is **an orthogonal direction to our work**.
>
> We want to first clarify that “silo-specific sample-level DP” is an important step away from the common “client-level DP” used in FL. This notion would directly apply to many real-world settings where we have prior knowledge that each data subject (person) corresponds to **at most one sample in a silo** (such as voting records across voting centers). I.e., **it is a special case** of the more general data-subject-level DP.
>
> More specifically, we believe that enforcing **general data-subject-level DP is still an active research area**; it may require **custom strategies for different scenarios**. For example,
> 1. if a subject has multiple records **within a silo**, then one intuitively expects the protection for that subject degrades gracefully under DP-SGD / sample-level DP (since we learn about that subject through different samples) and each silo can account for such cases independently (by group privacy; see also Appendix C.1). More broadly one may leverage results described in, e.g., recent works at NeurIPS’21 [d, e].
> 2. if a subject has records **across multiple silos** (a harder case), then in general one has to **detect the data subjects first** in order to provide data-subject-level DP. Techniques such as private set intersection (e.g. [a, b, c]) may be used, but when combined with DP, they may lead to privacy overhead (e.g. need for additional queries on top of training) and may require crypto protocols (e.g. [a, b]) which are orthogonal to our focus.
>
> Please also see footnote 2 (page 3) and Appendix C.1.
>
> We emphasize that our findings under our DP notion already raise important questions that also extend to data-subject-level DP and are worth further research. Developing techniques for enforcing general data-subject-level DP would be an interesting but orthogonal direction to this work.
>
> > …why letting each silo adds its own noise, instead of protecting the combined datasets (from all silos)
>
> The reviewer raised a very good point. In addition to our response above (challenges of enforcing general data-subject-level DP across silos), we point out that we are motivated by **how silos may realistically implement FL in practice (particularly with model personalization)**.
> - First, the nature of cross-silo FL often necessitates silos having their own **local, personalized models**; for example, if local datasets are large, simple local training may even be the ideal strategy (see, e.g., [f]), in which case silos do not communicate with others at all and thus should (and can only) keep their own DP guarantees for their models.
> - Second, even under the simplifying assumption that each data subject only contributes one sample in each silo (so sample-level DP suffices), computing an **accurate/tight** ($\varepsilon, \delta$) for sample-level DP over the **combined** dataset could be challenging **without trusted global statistics** (e.g. the sampling patterns across silos for privacy amplification theorems, the noise levels of each silo, or a centrally coordinated noise level); these statistics can be hard to curate if we want to maintain a trustless setting (L117). That is, while one *could* show that a global model (or the set of personalized models) satisfies some ($\varepsilon, \delta$)-DP using minimal information (e.g. with worst-case assumptions that data subjects appear in every silo and do privacy accounting using the minimum noise level across silos), the resulting guarantee could be quite loose & less meaningful.
> - Third, the computed ($\varepsilon, \delta$) over the combined dataset can also be less meaningful since silos can be **dynamic** (e.g. silos may exit the FL protocol at any time); silo-specific sample-level DP is directly compatible with **arbitrary client participation patterns** (as well as varying silo DP budgets).
>
> We believe that enforcing general data-subject-level DP, while ideal, is generally an open area as it also touches on other aspects of FL such as **client incentives** (e.g. whether a hospital would _want_ to spend resources to learn about / account for their patients’ appearances across _all other hospitals_). Moreover, we focus on private model personalization where **each client can keep its own model** and we believe that the reviewer’s suggestion would be more applicable if we only study the case where we have a **single shared model**.
>
> (continued)

---

> ### Author Response · Authors · 2022-08-02
> **Response to reviewer JfCt, part 2/2**
>
> (continued from part 1)
>
> ### Weakness #1 (cont.)
>
> > “item” vs “sample” in “silo-specific item-level DP”
>
> “Item” refers to a sample (e.g. an image) in a client’s local dataset, and we agree that changing to “sample” may be more clear. Thanks for this suggestion.
>
> > provide a formal DP definition of the silo-specific item-level privacy like the definition 2.1
>
> Thank you for the suggestion! Def 2.1 aims to provide a generic DP definition in terms of generic “records” and “datasets” and is thus not specific to a particular DP notion; if a “dataset” refers to a single client’s local samples, then each client has its own DP guarantees that are independent of all other clients’ DP guarantees (thus “silo-specific sample-level DP”). We agree that providing a definition can help the reader and will update accordingly.
>
>
> ### Weakness #2: “There is no novelty in the methodology, the paper only uses existing methods”
>
> We agree with the reviewer that we do not propose new algorithms (perhaps apart from the MR-MTL extension to handle cluster-structured heterogeneity at L270 and Appendix E.3). Please see our shared response above and we hope that the reviewer would consider the merits of this work in holistically studying and providing novel perspectives on the use of existing approaches for private cross-silo FL.
>
> ### Weakness #3: “The experiments only include two datasets, and performance difference is not very significant. It is unclear how solid the conclusion that MR-MTL is the best can be hold in general”
>
> We thank the reviewer for raising these issues. Regarding datasets, we point the reviewer to our response to shared concerns above.
>
> > “performance difference is not very significant”
>
> - We emphasize that we intended to highlight MR-MTL as **a strong baseline under this particular DP notion** due to its simplicity, extensibility (L270), minimal hyperparameters ($\lambda$), and minimal privacy overhead (L245) compared to existing personalization methods; **these properties make MR-MTL a tough-to-beat baseline, but not necessarily that it always has significantly superior performance**, since mean-regularization may not be optimal under arbitrary heterogeneity (e.g. Fig. 6). In fact, without adding DP, MR-MTL was shown in prior work (e.g. [h, i]) to be inferior to many existing methods that, e.g., leverage additional computation.
> - We also argue that the existence of a region of $\lambda$ values where it may outperform both endpoints of the personalization spectrum **under the same privacy budget** is an interesting phenomenon alone worth highlighting, particularly that **such utility advantage is flexible (depends on the privacy setting)** and can be made larger under privacy settings (Fig. 5, Propositions 6.5/6.6, L338-L347) compared to non-private settings.
> - See also our additional results on CIFAR-10, where the utility advantage of MR-MTL over local training / FedAvg can be large.
> - Note that all experiments are also repeated over multiple random seeds.
>
> ### Minor concerns
>
> > Define “local training”
>
> “Local training” refers to each silo **training and keeping their own model on their own local datasets, without any federated learning**; it is a very simple baseline that works surprisingly well in many cases; see, e.g., [f].
>
> > Fig.2, the right figure, explain what is meant by the semi-transparent plot line
>
> The semi-transparent lines refer to the local training / FedAvg baselines (reusing legend from Fig. 2 left) with the same $\varepsilon$ value as finetuning (so 0.5 in the right figure).
>
> ### Limitations
>
> Thank you for your thoughtful feedback; these are great questions. We have perhaps addressed Q1 and Q2 in Appendix C.1 along with our response to weakness #1 above and in Section 3 respectively, and we hope to add more discussions in the updated version.
>
> For Q3 (see also our response to weakness #1 above), we believe that the silo-specific sample-level DP notion would be directly applicable to settings where we have prior knowledge that each data subject (person) corresponds to at most one sample in a silo; real-world examples include
> - surgery records or single-shot vaccinations for a particular medical condition across hospitals (duplicates are unlikely);
> - voting records across counties/states for a particular election (a person can legally vote once); and
> - attendance records for a particular exam (e.g. entrance exams).
>
> We will add more discussion on this in the updated version.
>
> Refs:
> - [a] https://eprint.iacr.org/2017/299.pdf
> - [b] https://eprint.iacr.org/2017/799.pdf
> - [c] https://ieeexplore.ieee.org/stamp/stamp.jsp?arnumber=9343209
> - [d] https://arxiv.org/abs/2102.11845
> - [e] https://proceedings.neurips.cc/paper/2021/file/a89cf525e1d9f04d16ce31165e139a4b-Paper.pdf
> - [f] https://openreview.net/forum?id=GgM5DiAb6A2
> - [g] http://dimacs.rutgers.edu/~graham/pubs/papers/pdp.pdf
> - [h] https://arxiv.org/pdf/2003.13461.pdf
> - [i] https://arxiv.org/pdf/2012.08565.pdf

---

### Official Review · Reviewer_TNbE · 2022-07-18

**Rating:** 7
**Confidence:** 3
**Soundness:** 3 good
**Presentation:** 3 good
**Contribution:** 2 fair

**Summary:**

This paper considers the roles of privacy and personalization in cross-silo federated learning (FL) setting, which has less explored than in cross-device setting. This paper firstly shows why existing privacy solutions (employing differential privacy) are less appropriate to cross-silo FL with three key properties of silo-specific privacy. Then, it empirically shows that two notable phenomena under silo-specific item-level DP with two extreme baselines, local fine-tuning and federated average, which provides fundamental trade-off among privacy, utility and heterogeneity. It further shows mean-regularized multi-task learning (MR-MTL) is good interpolation between of the two extreme baselines while providing better trade-off between privacy and cross-silo data heteroneneity. Theoretical analysis of the MR-MTL is also included.


**Questions:**

Please explain more detailed reason that MR-MTL outperforms Ditto with a certain level of privacy budget ($epsilon=0.5 in Figure 5)

**Limitations:**

This paper has no negative societal impact as it studies the privacy protection in collaborative ML training.

**Strengths And Weaknesses:**

Overall, even though this paper does not provide new algorithm for cross-silo privacy protection, it raises the new notion of privacy suitable for cross-silo FL, and proposes interesting trade-off among privacy and data heterogeneity by exploring state-of-the-art personalized FL techniques. This paper is organized and written well, and hence it is easy to understand the difference between the notion of privacy in cross-device and cross-silo FL settings; characteristics of cross-silo privacy; and fundamental trade-off between privacy and heterogeneity in cross-silo FL with the strong baseline (MR-MTL). The role of the important hyper-parameter in the MR-MTL is well explored with empirical and theoretical analysis.

---

> ### Author Response · Authors · 2022-08-02
> **Response to reviewer TNbE**
>
>
> Thank you for taking the time to provide feedback, and we’re glad that the reviewer finds our work valuable! We provide our response below and please also see [our shared response](https://openreview.net/forum?id=Oq2bdIQQOIZ&noteId=rMtUPaS_fOR) for addressing concerns from other reviewers.
>
> > Explain why Ditto underperforms MR-MTL in Fig 5
>
> By construction, each client in Ditto requires at least two local training iterations (gradient steps) for updating its global and local model respectively (Line 5-6 of Algorithm 1 of [a]), whereas clients in MR-MTL require one iteration at a minimum (simply add a mean-regularization term to the batch gradient). Since the privacy cost under silo-specific item-level DP is directly correlated with the number of steps over the client’s local dataset, more iterations mean higher noise per iteration under the same total privacy budget. Fig. 5 aims to illustrate the effect of such privacy overhead.
>
> Alternatively one could also allow MR-MTL to run 2x many iterations (so as to match Ditto in terms of privacy cost), in which case the performance difference would come from faster/better convergence of MR-MTL.
>
> [a] https://arxiv.org/pdf/2012.04221.pdf

---

### Author Response · Authors · 2022-08-02
**Message to all reviewers**

We thank all reviewers for their time to provide helpful feedback for improving our paper! In addition to specific responses to each reviewer, we’d like to provide a summary and address some shared comments.

First, we are glad that reviewers find our problem of interest relevant (TNbE, JfCt, 7hbo), the privacy notion for cross-silo FL and our observations interesting (TNbE, 7hbo, dzK1), our empirical/theoretical analysis useful for understanding the behavior of MR-MTL (TNbE, 7hbo), that our work provides novel perspective on private cross-silo learning (TNbE, 7hbo), and that the paper is generally well written and presented (TNbE, dzK1, 7hbo)

We now address some shared comments from the reviewers below.

**[Adding more experimental evaluation (JfCt, dzK1)]**

Following the reviewers’ suggestions, **we added results on heterogeneous CIFAR-10** with a setup following previous work (e.g. [a,b]) **at the beginning of the updated Appendix in green in the ZIP file** (will move to the main text in the updated version). We provide both:
- The privacy-utility tradeoff curves (similar to Fig. 3), where MR-MTL is consistently competitive, and
- Interpolation curves with varying $\lambda$ (similar to Fig. 5), where we observe behaviors of MR-MTL similar to those seen in Fig. 5, such as that the best $\lambda^*$ becomes larger under stronger privacy (Thm 6.3) and the utility curve as a function of $\lambda$ has a (roughly) quasi-concave (“bump”) shape (Lemma 6.7).

We’d also like to point out that this is the 6th dataset on top of the 3 datasets in the main text and 2 datasets in the Appendix. We may also expand to more datasets in the updated version to further strengthen the evaluation.

**[Concerns about limited algorithmic novelty (JfCt, dzK1)]**

We’d like to clarify that our goal is not to propose a new algorithm but to comprehensively examine private cross-silo federated learning from different perspectives:
1. What DP notions may be more suitable/practical than the commonly used client-level DP?
2. For a candidate DP notion, what are some emerging phenomena that may go against intuitions developed from client-level DP?
3. What are some good baselines under this DP notion, and why might a particular baseline work better than others?
4. Having identified strong baselines, what are then some limitations or challenges of deploying them?

To study these questions, our work:
1. explores silos-specific item(sample)-level DP as a realistic alternative to client-level DP for cross-silo FL;
2. studies the emerging phenomena related to privacy, heterogeneity, and personalization;
3. dissects the desiderata of a strong baseline and how existing methods may fall short in terms of privacy overhead (Fig. 4, L229, Appendix D.3, etc.);
4. characterizes how MR-MTL as one of the simplest & strongest baselines interacts with privacy and heterogeneity; and
5. examines the complications of actually leveraging personalization methods to reconcile with the emerging phenomena, through the lens of private hyperparameter tuning (sec. 7).

Overall, we hope that our work serves to shed more light on private cross-silo FL with both positive and negative results and spur future efforts in this area. We also provide additional results and discussions in the appendix that could be useful for future work.

Moreover, we believe that model personalization under privacy is in general an important and active research area and that as reviewers TNbE and 7hbo pointed out, providing novel perspectives – even with existing algorithms – could be valuable to the community.

Refs:
- [a] http://proceedings.mlr.press/v139/shamsian21a/shamsian21a.pdf
- [b] https://proceedings.neurips.cc/paper/2020/file/f4f1f13c8289ac1b1ee0ff176b56fc60-Paper.pdf

### Note on the updated appendix, and not updating the main paper
Due to page limit, we keep our responses to the reviewers’ feedback on OpenReview and do not update the main paper during the rebuttal; however, **we updated the top of the Appendix (in ZIP file) to provide additional results**. We will incorporate all feedback in the updated version.

---

### Author Response · Authors · 2022-08-07
**Gentle follow-up with reviewers**

Dear reviewers,

Thank you again for taking the time to provide valuable feedback! We’d like to follow up with you and check if our responses/additional results have adequately resolved your questions around the privacy notion/definition (Reviewer JfCt, 7hbo), algorithmic novelty (Reviewer JfCt, dzK1), empirical evaluation (Reviewer JfCt, dzK1), and writing/presentation clarity (Reviewer 7hbo). If not, we'd really appreciate it if you could let us know if there’s anything we could discuss and/or clarify.

Thank you,

Paper3693 Authors

---

### Author Response · Authors · 2022-08-08
**Another gentle follow-up with reviewers**

Hi reviewers,

Thanks again for your valuable feedback! Since the author-reviewer discussion period ends in 24hrs, we’d like to kindly follow up with you again to check if our responses/additional results have adequately resolved your questions. We'd love to know if there’s anything else we could discuss and/or help clarify.

Thank you,

Paper3693 Authors

---

### Meta-Review · Area_Chair_N39K · 2022-08-24

**Recommendation:** Accept
**Confidence:** Less certain

**Metareview:**

The paper presents an analysis of item-level or sample-level DP with personalization in cross-silo federated learning.

The reviews are strongly divided with two recommending acceptance and two recommending rejection.

After reading all the reviews and the paper, I find the argument of the "accept" side stronger.

While the paper does not propose new algorithms, it presents a systematic analysis of existing methods that helps explain their properties. I believe this can be a more valuable contribution to the community than yet-another-new-algorithm.

That said, there are also important weaknesses, as noted by the reviewers. It is not clear how to select the optimal $\lambda$, especially in the non-convex setting with no theory as a guide. It is therefore not clear how useful the method would be in actual application where suitable $\lambda$ would have to be found somehow. This seems like an obvious topic of future work. A broader assessment with more data sets and some actual algorithm to select $\lambda$ would clearly strengthen the paper.

Some reviewers were confused by the proposed definition for sample-level cross-silo DP. I would strongly encourage the authors to use a part of the extra space for camera ready for writing this definition explicitly to avoid such confusion.

**Award:**

No

---

### Decision · Program_Chairs · 2022-09-14

Accept